**Technical Report**

# Single duplex DNA sequencing with CODEC detects mutations with high sensitivity

Jin H. Bae [1,9], Ruolin Liu [1,9], Eugenia Roberts[1], Erica Nguyen[1], Shervin Tabrizi [1,2,3], Justin Rhoades[1], Timothy Blewett[1], Kan Xiong[1], Gregory Gydush[1], Douglas Shea[1], Zhenyi An[1], Sahil Patel[1,2,3], Ju Cheng[1], Sainetra Sridhar[1], Mei Hong Liu[4], Emilie Lassen [5], Anne-Bine Skytte [5], Marta Grońska-Pęski [4], Jonathan E. Shoag[6], Gilad D. Evrony[4], Heather A. Parsons [7], Erica L. Mayer[7], G. Mike Makrigiorgos[7], Todd R. Golub [1,7,8] & Viktor A. Adalsteinsson [1]✉

Detecting mutations from single DNA molecules is crucial in many fields but challenging. Next-generation sequencing (NGS) affords tremendous throughput but cannot directly sequence double-stranded DNA molecules ('single duplexes') to discern the true mutations on both strands. Here we present Concatenating Original Duplex for Error Correction (CODEC), which confers single duplex resolution to NGS. CODEC affords 1,000-fold higher accuracy than NGS, using up to 100-fold fewer reads than duplex sequencing. CODEC revealed mutation frequencies of $2.72 \times 10^{-8}$ in sperm of a 39-year-old individual, and somatic mutations acquired with age in blood cells. CODEC detected genome-wide, clonal hematopoiesis mutations from single DNA molecules, single mutated duplexes from tumor genomes and liquid biopsies, microsatellite instability with 10-fold greater sensitivity and mutational signatures, and specific tumor mutations with up to 100-fold fewer reads. CODEC enables more precise genetic testing and reveals biologically significant mutations, which are commonly obscured by NGS errors.

Discovering extremely low-abundance mutations as rare as within a single double-stranded DNA molecule (a 'single duplex') is crucial to finding diagnostic[1,2], predictive[3,4] and prognostic[5,6] biomarkers; understanding cancer evolution[7,8] and somatic mosaicism[9,10]; and studying infectious diseases[11,12] and aging[13,14]. In principle, single-molecule sequencing technologies (for example, PacBio and Oxford Nanopore Technologies) can keep single DNA duplexes intact throughout their workflows to sequence them in whole to resolve true mutations on both strands apart from false mutations on either strand. However, in practice, they lack the required accuracy and throughput[15,16].

Next-generation sequencing (NGS), on the other hand, continues to offer superior read accuracy and throughput[17] but is not configured to sequence single duplexes—at least not without severely compromising its throughput or utility.

NGS affords high throughput by reading short, clonally amplified DNA fragments in massively parallel fluorescence analysis. Yet, its accuracy is limited by the need to dissociate Watson and Crick strands of each DNA duplex. Without a complementary strand for comparison, errors introduced on either strand due to base damage, PCR and sequencing[18] can be disguised as real mutations. While it is possible to

[1]Broad Institute of MIT and Harvard, Cambridge, MA, USA. [2]Koch Institute for Integrative Cancer Research at MIT, Cambridge, MA, USA. [3]Massachusetts General Hospital, Boston, MA, USA. [4]Center for Human Genetics and Genomics, Departments of Pediatrics and Neuroscience & Physiology, New York University Grossman School of Medicine, New York City, NY, USA. [5]Cryos International, Aarhus, Denmark. [6]University Hospitals Cleveland Medical Center, Case Western Reserve University School of Medicine, Case Comprehensive Cancer Center, Cleveland, OH, USA. [7]Dana-Farber Cancer Institute, Boston, MA, USA. [8]Harvard Medical School, Boston, MA, USA. [9]These authors contributed equally: Jin H. Bae, Ruolin Liu. ✉e-mail: viktor@broadinstitute.org

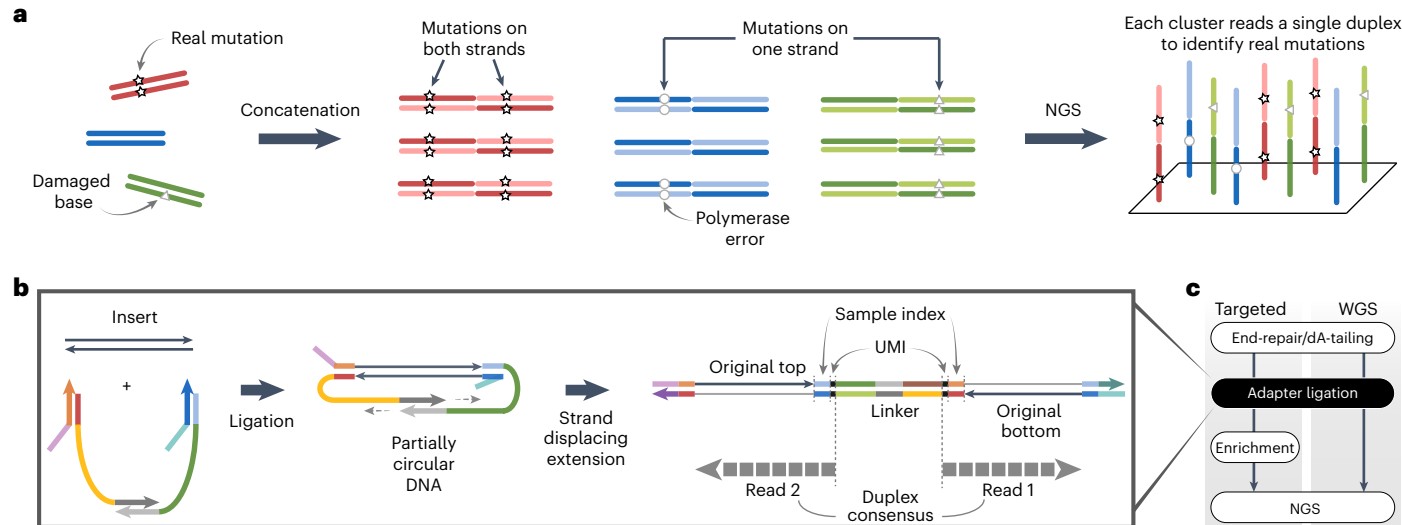

**Fig. 1 | Overview of CODEC. a**, To distinguish real mutations from damaged bases or polymerase errors, CODEC physically links Watson and Crick strands of each original duplex, which may have an alteration confined to one strand. Each cluster reads an NGS library molecule with sequences of both strands to trace a whole duplex, hence single duplex sequencing. **b**, CODEC uses the adapter quadruplex, which is prepackaged with all of the components needed for Illumina NGS, followed by strand displacing extension. Unlike standard NGS, CODEC can read outward to sequence a UMI, an index and an insert together. Each NGS read pair becomes self-sufficient for forming a consensus between two strands of an original duplex. **c**, CODEC is compatible with both targeted sequencing and WGS.

use unique molecular identifiers (UMIs) to separately track both strands of each DNA molecule to detect true mutations[19], this does not solve the underlying limitation of NGS—duplex dissociation. For example, duplex sequencing[20], which has been the gold standard of high-accuracy sequencing and used by other recent methods[21,22], tags double-stranded UMIs on each original duplex to achieve >1,000-fold higher accuracy, but recovering both strands among many other strands could require 100-fold excess reads[23], which severely limits its utility.

To date, several methods have sought to overcome the low efficiency of generating a duplex consensus, but they are limited to either targeted panels or shallow sequencing. Duplex proximity sequencing (Pro-Seq)[24] and SaferSeqS[25] use multiplexed PCR, limiting their applications to small targeted panels with deep sequencing. BotSeqS[21] performs 10^5-fold sample dilution to increase the chance of recovering both strands, which is not compatible with targeted deep sequencing or applications that require high sensitivity. NanoSeq[22] improved the accuracy of BotSeqS by pairing it with a restrictive end-repair method that uses dideoxy bases to avoid nick extension, but as a result, it is not compatible with highly fragmented samples such as cell-free DNA (cfDNA) and is limited to 29% of the human genome when using its standard protocol. Some technologies such as o2n-seq[26] and circle sequencing[27] only use a single strand of a duplex, and thus lack the ability to create a duplex consensus. Despite the need for sequencing duplexes with high accuracy and throughput, there have only been methods for limited applications.

We thus reasoned that linking the information of both strands before strand dissociation could make NGS capable of reading single DNA duplexes with high efficiency. Here we developed a hybrid method called Concatenating Original Duplex for Error Correction (CODEC) that combines the massively parallel nature of NGS and the resolution of single-molecule sequencing by reading both strands of each DNA duplex with single NGS read pairs (Fig. 1a). Any differences between concatenated sequences would indicate alterations confined to one strand from either noncanonical base pairing created by nucleobase damage or an error introduced during PCR amplification or sequencing. CODEC is compatible with major NGS workflows ranging from targeted sequencing to whole-genome sequencing (WGS).

## Results

### Adapter quadruplex and workflow

The CODEC structure can be built by replacing a typical adapter duplex with the CODEC adapter quadruplex, containing all elements required for NGS. We rationally designed double-stranded segments of the adapter to hold the whole quadruplex and introduced single-stranded segments to mitigate the bending stiffness of the DNA double helix (Extended Data Fig. 1a,b). After adapter ligation seals both ends of an input molecule, strand displacing extension initiates at the remaining 3′-ends to elongate each strand by using the opposite strand as a template (Fig. 1b). This strategy allowed CODEC to physically concatenate the Watson strand with the reverse complement of the Crick strand into a single strand without forming a prohibitive hairpin or inverted repeat structure from two complementary sequences. The resulting structure is two original strands concatenated with the CODEC linker in the middle and Illumina adapters on both sides. The molecular process depicted in Fig. 1b replaces the adapter ligation step of commercial NGS library preparation workflows (Fig. 1c).

To fully use the concatenated structure, we also relocated the NGS library components. In contrast to the conventional Illumina structure with the NGS read primer binding sites on the outer side, we moved the binding sites to the CODEC linker and sequenced outward to prevent reading byproducts without the linker and improve quality scores (Extended Data Fig. 1c–e). Sample indices, which are typically added after adapter ligation and read separately from the inserts, were added during the ligation and read together with the inserts. This structure further suppressed index hopping compared to using typical unique dual indices[28] (0.056% versus 0.16%). We designed sets of four sample indices that have all four bases at every position to ensure a high base diversity necessary for image analysis on Illumina sequencers (Extended Data Fig. 1f).

### Proof of concept

We first confirmed that the CODEC workflow could create the intended NGS library structure by performing CODEC on four human cfDNA samples. We found that 72.8% of the reads showed the correct structure and retained information from both strands (Extended Data

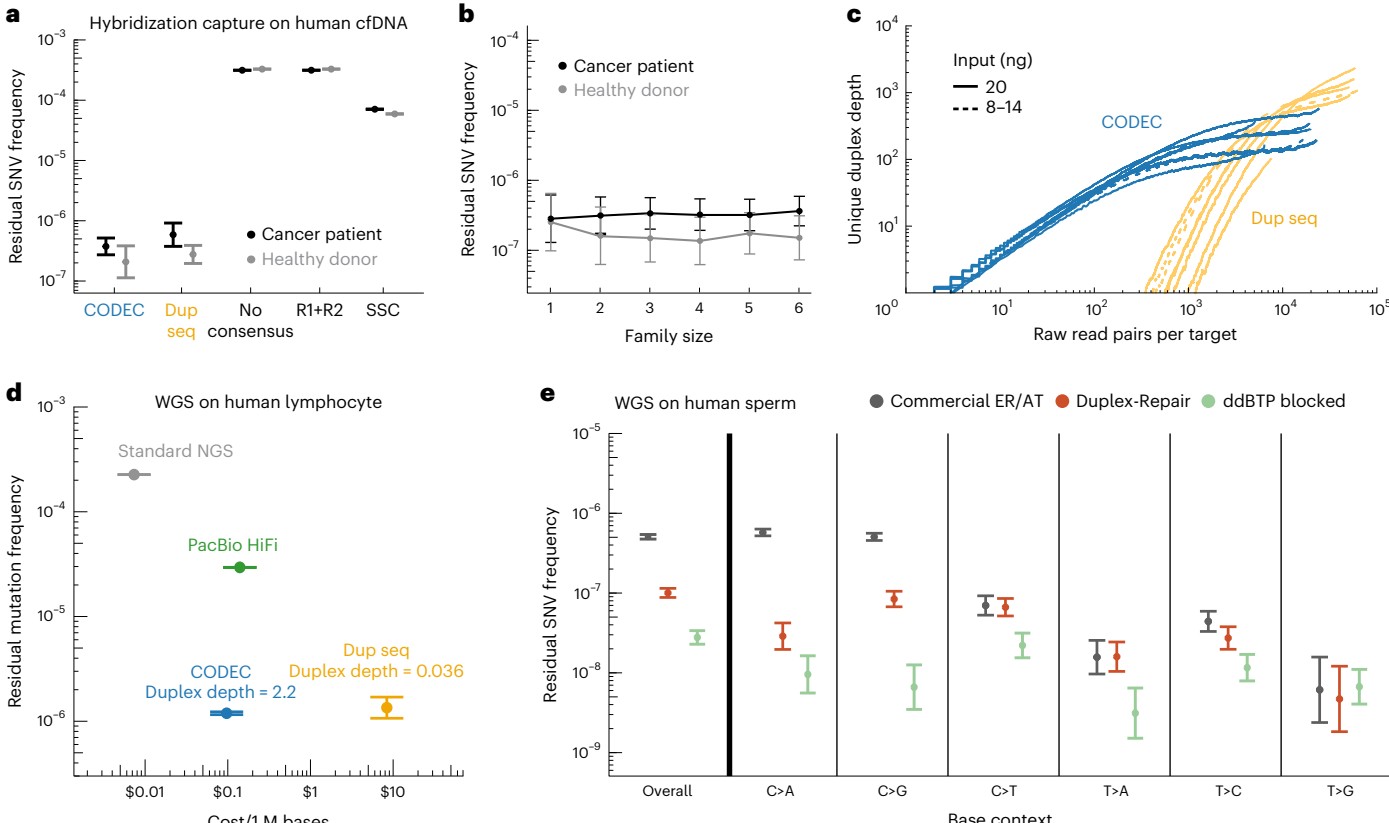

**Fig. 2 | Proof of concept. a**, Residual SNV frequencies of CODEC, duplex sequencing and other consensus methods such as paired-end reads consensus (R1 + R2) and SSC. Target enrichment with a pan-cancer gene panel was performed on cfDNA of two individuals. Duplex sequencing required at least two reads of each strand[22]. **b**, CODEC residual SNV frequencies at each family size, which is the number of raw read pairs with the same UMI and start-stop positions. **c**, Recovery of unique original duplexes per targeted site in cfDNA of cancer patients and healthy donors against raw read pairs per target. **d**, Residual mutation frequencies and sequencing costs of different methods for WGS of the pilot genome NA12878 of the Genome in a Bottle Consortium. Because duplex sequencing WGS could not recover any duplex with the standard threshold, we had to relax it to one read of each strand only for this analysis. **e**, Residual SNV frequencies of WGS on human sperm with different end-repair/dA-tailing methods. **a,b,d,e**, Data points and error bars indicate mean values and 95% binomial confidence intervals by Wilson method, respectively.

Fig. 2a–e). Because each insert does not need to be fully sequenced to determine the structure (Supplementary Text), the ratio of correct reads was not negatively affected by the longer insert size (Extended Data Fig. 2f). More details on each NGS dataset can be found in Supplementary Table 1.

For a fair comparison throughout this work, we matched CODEC and standard Illumina NGS on raw read pairs and deduplicated sequencing coverage for targeted and WGS, respectively, as per standard conventions (Supplementary Table 2)[29]. For CODEC, deduplicated coverage includes all byproducts (Extended Data Fig. 2a) that do not pass all CODEC filters, representing the cost of the sequencing. Unless otherwise indicated, coverage (for example, 30×) refers to deduplicated coverage, whereas correct product depths are calculated based on deduplicated CODEC reads with the correct structure. Duplex depths are computed after all CODEC filters are implemented. Thus, duplex depths for CODEC will always be lower than the indicated coverages. However, all raw or deduplicated CODEC reads can still be used for standard NGS analyses such as clonal mutation detection with established tools.

We next explored whether the fragments with the correct structure could provide comparable error rates to duplex sequencing using substantially fewer reads. To meet the high sequencing coverage requirements of duplex sequencing, we ran target enrichment with the pan-cancer hybridization capture panel (800 kb) on NGS libraries, constructed from 20 ng cfDNA of a cancer patient and a healthy

donor. Of note, sequenced bases that are different from the reference were named 'residual single-nucleotide variant (SNV)' or 'residual indel' instead of 'error' to reflect the low-abundance mutations left after excluding the germline and high-abundance (for example, >1% variant allele frequency (VAF)) somatic mutations[20,22,30]. When compared to the ground truth based on high-abundance mutations from duplex sequencing data, we found that the mean CODEC residual SNV frequency of two individuals ($2.9 \times 10^{-7}$) was similar to that of duplex sequencing ($4.3 \times 10^{-7}$; Fig. 2a). Because error rates are affected by multiple factors other than a sequencing technology itself, we used the same experimental and computational protocols for all methods whenever applicable. The minimum required reads of each strand were two for duplex sequencing[22,23]. As expected, a consensus of Watson and Crick strands was crucial for suppressing errors; although single-strand consensus (SSC) among all Watson strand reads or all Crick strand reads (Extended Data Fig. 3) was more accurate than the no consensus and paired-end reads consensus (R1 + R2), residual mutation frequency of CODEC was 234-fold lower than that of SSC. Notably, even a single read pair of CODEC was equally accurate (Fig. 2b), as each of them is self-sufficient to form a duplex consensus.

CODEC had a strong advantage in the number of reads required to uncover the same number of unique DNA duplexes. When we sequenced more cfDNA samples of healthy donors and breast cancer patients (CODEC median 25 M raw read pairs, duplex sequencing median 31 M), we found that duplex sequencing could not start

recovering original duplexes until receiving 600 read pairs per target on average (Fig. 2c). In contrast, CODEC started to recover them with 220-fold fewer read pairs despite byproducts and off-target reads. The gap between required reads was maximized when recovering a smaller number of duplexes, suggesting that CODEC could be uniquely capable of sequencing broad genomic regions with shallow coverage. On the other hand, duplex sequencing eventually recovered more unique duplexes when it obtained enough read pairs (Extended Data Fig. 4a,b), although each duplex sequencing library needs different amounts of sequencing depending on its molecular complexity[21]. The lower endpoints of CODEC (mean 164 duplexes versus duplex sequencing mean 466 duplexes) were probably due to the lower library conversion efficiency of CODEC.

We next sought to determine whether CODEC could enable human whole-genome 'duplex' sequencing, which would otherwise be impractical due to high cost. To assess this, we applied CODEC (214 M raw read pairs) and duplex sequencing (305 M) to WGS of the pilot genome NA12878 of the Genome in a Bottle Consortium[31]. Because duplex sequencing WGS could not recover any duplexes with the standard threshold, we had to relax it to one read of each strand only for this analysis (Extended Data Fig. 4c). Throughout this work, we used the same variant calling pipeline for CODEC and standard WGS unless otherwise noted (see 'CODEC single-fragment mutation caller' in Methods). In a cost-benefit analysis, the cost of CODEC was 87 times lower than that of duplex sequencing while maintaining higher accuracy than standard WGS (Fig. 2d), although CODEC WGS was not as uniform as that of standard WGS at high GC content (Extended Data Fig. 5a,b). Due to the high cost of duplex sequencing, we compared CODEC with standard WGS after this point and only used duplex sequencing for targeted panels. In addition, CODEC showed similarly low residual SNV frequencies in whole-exome sequencing (WES) of human genomic DNA (Extended Data Fig. 5c).

We reasoned that some errors in CODEC sequencing resulted from end-repair/dA-tailing (ER/AT) as shown for duplex consensus[22,32]. Indeed, residual mutation frequencies in CODEC were generally higher toward the ends of DNA fragments (Extended Data Fig. 6a). To address these issues, we paired CODEC with varied ER/AT methods and applied them to sperm DNA with an expectedly low biological mutation rate. We compared a commercial ER/AT method with Duplex-Repair[32] and a custom ddBTP-blocked ER/AT inspired by ref. 22, which fully blocks ER/AT errors in theory (Extended Data Fig. 6b). When we applied these to a sperm DNA (39-year-old donor) before CODEC, Duplex-Repair and ddBTP-blocked ER/AT showed 5.1-fold and 18.6-fold lower residual SNV frequencies ($1.00 \times 10^{-7}$ and $2.72 \times 10^{-8}$) than a commercial ER/AT kit ($5.07 \times 10^{-7}$), respectively (Fig. 2e and Extended Data Fig. 6c). With ddBTP-blocked ER/AT, the result was comparable to that of the recent report ($1.48 \times 10^{-8}$ for 21 year old and $4.38 \times 10^{-8}$ for 73 year old)[22].

### Detection of germline and somatic mutations

Given the superior accuracy and single duplex resolution of CODEC, we then explored the various types of biological analyses that could be uniquely enabled, starting with germline genetic testing. Traditionally, with standard NGS, a mutation caller is required to discern true mutations from errors based on their evidence in multiple independent reads (Fig. 3a). To test germline mutation detection at low coverage, we used an established caller instead of the single-fragment mutation caller (see 'Germline SNV and small indel calling in downsampled WGS' in Methods). When we compared CODEC and standard WGS of NA12878 at coverages ranging from 1× to 5× (0.6–3.0× correct product depth), CODEC showed 21-fold fewer false positives (FP) and 2-fold more false negatives (FN) for germline SNPs than standard WGS, which was mostly caused by byproducts (Fig. 3b). At 40× coverage (17× correct product depth), the FN rate was further reduced to 0.057. The best FN (0.026) was achieved when also including nonduplex reads such as the byproducts (Supplementary Table 3). This result

implied that CODEC WGS may better resolve low-abundance mutations, which are obscured by FP in standard WGS, especially with shallow sequencing coverage.

To examine the potential for CODEC to detect low-abundance mutations, such as those arising from clonal hematopoiesis (CH), directly from single DNA duplexes, we analyzed 6× CODEC WGS (0.47–1.02× duplex depth) after Duplex-Repair and 6× standard WGS on buffy coat-derived germline DNA from 15 breast cancer patients. Duplex-Repair was not applied to standard WGS because its effect was negligible on standard Illumina NGS[32]. As expected, only CODEC was able to reveal a linear relationship between the number of somatic mutations and age ($R^2 = 0.80$; Fig. 3c). The number of somatic mutations acquired per year in mature white blood cells was similar to a recent report (20.2 versus 19.8; ref. 22).

For cross-validation of the single duplex mutations, we also performed CODEC WGS and targeted deep sequencing using newly created duplex sequencing libraries from buffy coat DNA of eight breast cancer patients. An average of 8.2% (range: 2.7–14.6%) of the mutations with their VAF down to 0.012% were observed again by sequencing them to a median of 2,311 (range: 225–6,970) duplex depth, implying that they were real somatic mutations (Extended Data Fig. 7a). We estimate that more could be found to be true somatic mutations at lower abundance if we were to sequence even deeper (Extended Data Fig. 7b).

We then used a similar cross-validation strategy to test if observing genome-wide CH from cfDNA is possible. Targeting the single duplex mutations detected by 4.3× (0.47–0.83× duplex depth) CODEC WGS on cfDNA of four healthy donors and four breast cancer patients, we performed deep duplex sequencing on matching buffy coat DNA to confirm their origin. On average, 4.6% and 9.5% of the single duplex mutations detected in cfDNA were also present in buffy coat DNA of healthy donors and patients, respectively, with their observed VAF as low as 0.0052% and the median duplex depth of 14,529 (range: 11,174–16,567; Fig. 3d). Standard Illumina NGS with >100-fold higher residual mutation frequency would have resulted in only <0.05–0.1% of mutations targeted being true mutations. Trinucleotide context (TNC), that is, a mutation as well as their neighboring nucleotides, of the validated mutations was different for each individual except high peaks at C > T mutations at CpG as expected[33] (Fig. 3e) with no errors specific to CODEC (Extended Data Fig. 7c). Our results suggest that CODEC could be uniquely capable of detecting rare somatic mutations in white blood cells and cfDNA such as those associated with CH using minimal sequencing.

### Enhanced tumor mutation profiling

Given the ability to detect both high- and low-abundance mutations with fewer reads, we next sought to determine what impact this could have on cancer genome sequencing. To confirm that detecting tumor mutations with low-coverage WGS is possible with CODEC but not standard WGS, we compared them at 2× coverage by testing how well they detect mutations (VAF > 1%) established by deep standard WGS (60× coverage) paired with a variant caller, Mutect2 (ref. 34), on eight breast tumor samples. With higher accuracy, 2× CODEC (0.11–0.14× duplex depth) had 83 times higher fraction of SNV validated by deep standard WGS + Mutect2 than 2× standard WGS (Fig. 4a and Extended Data Fig. 8a), and increasing CODEC coverage improved sensitivity as expected (Fig. 4b). Of note, deep CODEC WGS did not outperform deep (for example, 60×) standard WGS in its sensitivity for high-abundance (for example, clonal) tumor mutations, which was expected, as this is driven mostly by NGS depth, not accuracy. That said, we saw high positive predictive values (PPVs) of mutations detected in single duplex reads from deep CODEC WGS, suggesting that single CODEC reads are accurate enough to detect true tumor mutations. To further explore this, we performed targeted resequencing of mutations identified exclusively in single duplexes of 2× CODEC WGS and found more than a quarter to be validated as true mutations (Fig. 4c). The portion

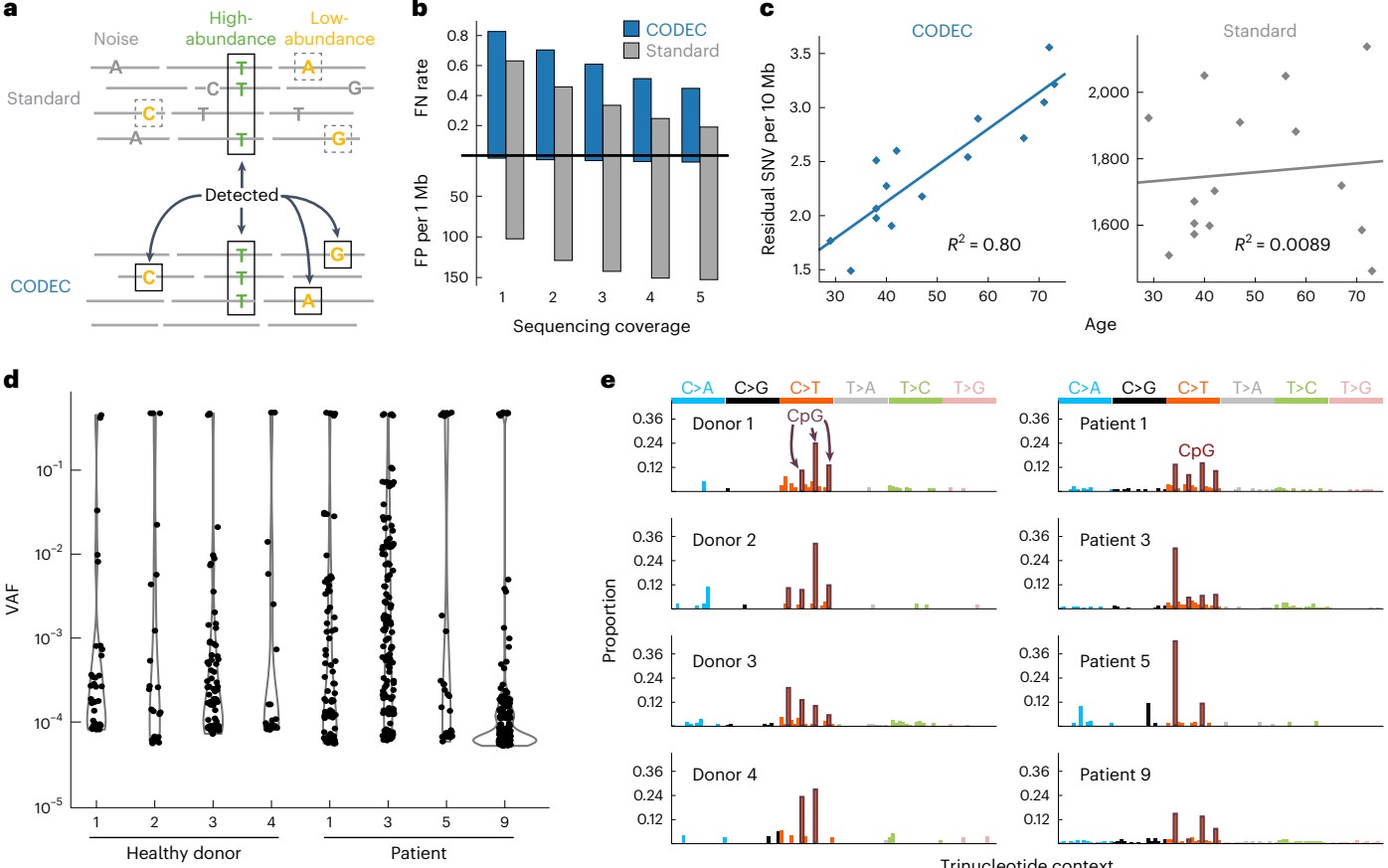

**Fig. 3 | Detection of rare mutations enabled by high sequencing accuracy.** **a**, Standard NGS can only detect high-abundance mutations detected in multiple molecules but not low-abundance mutations that are obscured by background noise. CODEC can detect both high- and low-abundance mutations due to its single duplex resolution. **b**, FP and FN of CODEC and standard WGS on NA12878 when downsampled to 1× to 5× (0.6–3.0× correct product depth). **c**, Residual somatic SNVs detected from 6× CODEC (0.47–1.02× duplex depth) paired with duplex repair and 6× standard WGS on buffy coat DNA of 15 breast cancer patients. The fitted lines and $R^2$ values were from linear regression. **d**, VAF of the genome-wide mutations arising from CH discovered by CODEC (0.47–0.83× duplex depth) on cfDNA and independently validated by duplex sequencing in buffy coat. **e**, Mutation spectra of genome-wide CH from **d**. Each bar represents a TNC.

of mutations exclusively identified by CODEC showed VAF down to 0.042%. In contrast, mutations that were also detected by Mutect2 were limited to VAF > 1%. These results suggest that low-coverage CODEC can detect rare mutations that cannot be discovered by standard WGS even with deep coverage.

We reasoned that such capability may enable analyzing mutation sequence contexts across the whole genome without deep sequencing. To test the idea, we compared CODEC and standard WGS with the single-fragment mutation caller to standard WGS with Mutect2 and investigated differences in TNC of mutations in a colon tumor sample with microsatellite instability (MSI). Analyzing all mutations from standard WGS (12× coverage) resulted in a different profile compared to selecting only mutations detected by Mutect2, suggested by low cosine similarity of 0.61 (Fig. 4d). In contrast, accepting all mutations from 5× CODEC generated a profile similar to that of using high-abundance mutations with cosine similarity of 0.98. The same trend was consistently observed at even lower CODEC sequencing coverages. When we compared mutation profiles between high-abundance mutations selected from standard WGS and its downsampled data, lowering coverage below 7× started to reduce cosine similarity (Fig. 4e). CODEC successfully maintained high cosine similarity over 0.9 (ref. 35) against high-abundance mutations down to 0.05×, reducing the sequencing coverage required by 140-fold, or reads by 105-fold, to acquire the same mutation profile. This implies that low-coverage CODEC identified both low- and high-abundance mutations reliably.

We then extracted Catalogue Of Somatic Mutations In Cancer (COSMIC) signatures[36] from the mutation profiles of the MSI patient (Fig. 4f). CODEC detected not only both MSI signatures observed among high-abundance mutations selected by Mutect2 but also one more MSI signature, SBS21. Using all mutations from standard WGS canceled most of the MSI signatures. Signatures of mutations detected by CODEC but discarded by Mutect2 resembled those of all mutations from CODEC, suggesting that they were likely from real somatic mutations as well.

To test whether low-coverage CODEC could determine clinically significant tumor characteristics, we expanded the mutational signature analysis to eight breast cancer patients. Cosine similarity between mutation profiles of 2× CODEC and 60× standard WGS paired with Mutect2 again remained high, whereas that of 2× standard WGS did not (Fig. 4g). When the mutation profiles were further analyzed for COSMIC signature 3 that implies homologous recombination deficiency (HRD) status of a patient, which is a frequent driver of tumorigenesis[37] and a predictive biomarker[38], correlation between weights of signature 3 among all signatures estimated by CODEC and Mutect2 resulted in Pearson's coefficient = 0.91 (Fig. 4h and Extended Data Fig. 8b). This was higher than −0.12 from the same calculation using standard WGS. In addition, CODEC determined all positive and negative HRD statuses correctly, according to CHORD[39] and Mutect2.

Another potential application of CODEC is tracking mutations of interest with fewer reads. To test how CODEC improves tumor mutation detection from liquid biopsy samples, we performed hybridization

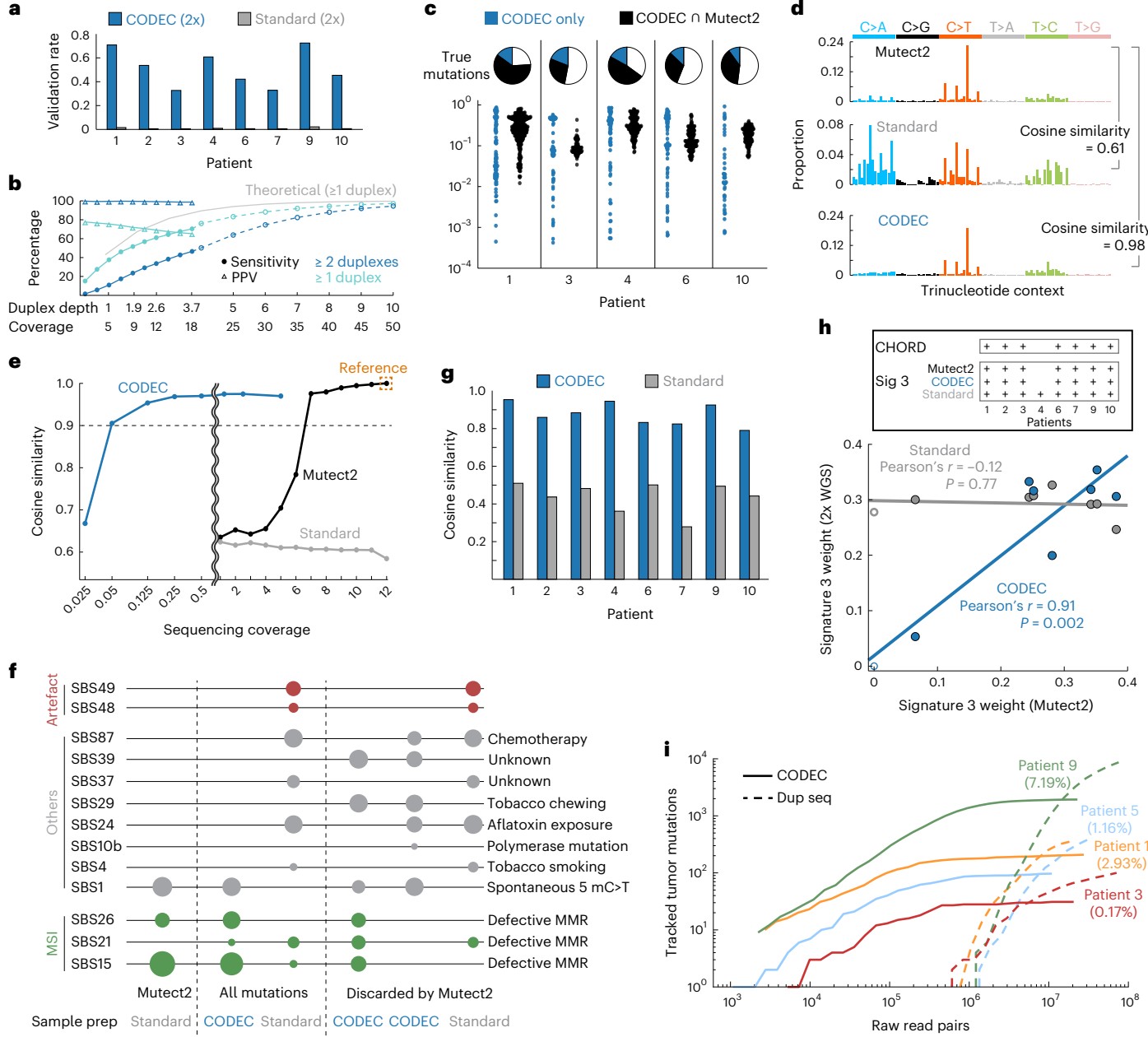

**Fig. 4 | Detection of somatic mutations in cancer genomes. a**, Mutations detected by 2× CODEC (0.11–0.14× duplex depth) and standard WGS on eight breast tumor samples were validated by 60× standard WGS + Mutect2. Mutations on a single read were accepted. **b**, Sensitivity for detecting clonal tumor mutations from CODEC WGS (3× duplex depth; 18× coverage) and corresponding PPV. The data were analyzed by requiring either ≥1 or ≥2 duplexes bearing the same mutation. Standard Illumina NGS at ×18 coverage paired with Mutect2 (ref. 34) achieved 98.6% sensitivity and 92.8% PPV. Theoretical numbers and projections (dashed) were calculated based on binomial models. **c**, Ratios and VAF of mutations initially found in CODEC WGS on tumor samples, cross-validated by targeted duplex sequencing on the same sample. Mutations exclusively detected by CODEC were grouped separately. **d**, Mutation spectra from 12× standard WGS with or without Mutect2 and 5× CODEC (2× duplex depth)

data of a colon tumor with MSI. **e**, Cosine similarities against high-abundance mutations selected by Mutect2 from 12× standard WGS (orange box). Each method was downsampled to lower coverages. **f**, COSMIC signatures extracted from different categories of mutations. Categories under 'discarded by Mutect2' are subsets of corresponding categories under 'All mutations'. **g**, Cosine similarities between 60× WGS + Mutect2 and either 2× CODEC or standard WGS on eight breast tumor samples. **h**, Pearson correlation between weights of HRD signature 3 estimated by 60× WGS + Mutect2 and either 2× CODEC or standard WGS. *P* values were calculated from simple linear regression with null hypothesis that slope is 0. The box shows the ground truth of HRD statuses determined by CHORD. **i**, Tumor mutations of four breast cancer patients tracked by performing hybridization capture on their cfDNA with personalized probe panels. Percentages indicate tumor fractions.

capture with personalized, tumor-informed probe panels[23] on both CODEC (median: 24 M raw read pairs) and duplex sequencing (median: 52 M) libraries from four breast cancer patients' cfDNA. Four patients with different tumor fractions showed up to a 100-fold reduction in read pairs needed to detect the tracked mutations (Fig. 4i and Supplementary Table 4).

## Improved detection of MSI

Considering that CODEC WGS showed 290-fold lower residual indel frequencies than standard WGS (Extended Data Fig. 9) and that Illumina NGS is known to have an especially high indel error rate at DNA homopolymers, we explored if CODEC might improve MSI detection, which is an FDA-approved indication for immunotherapy

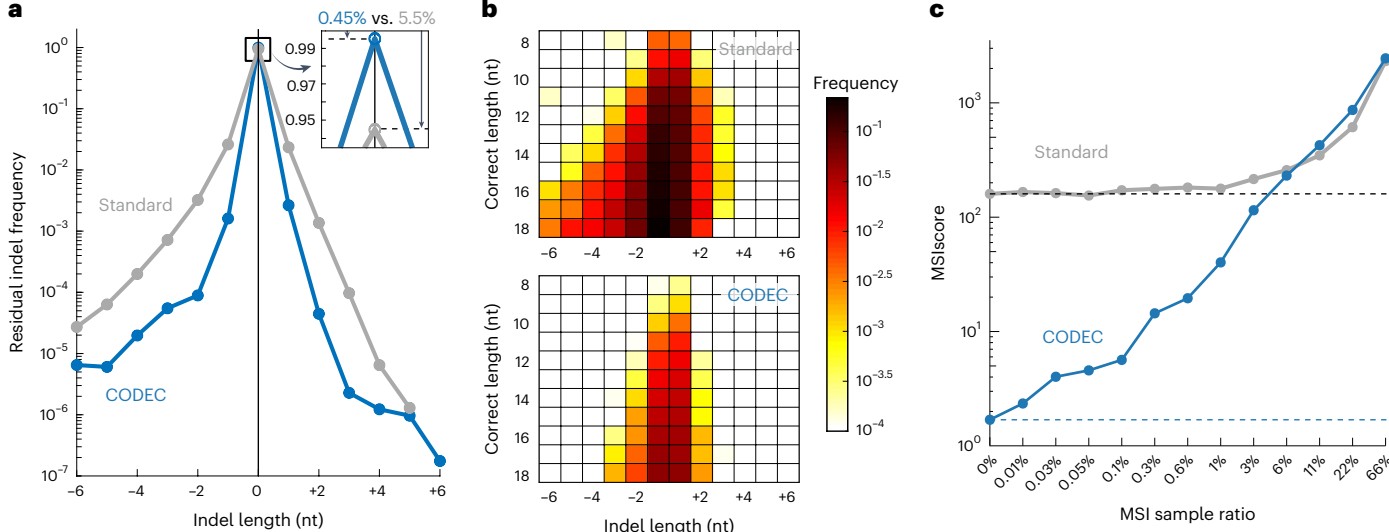

**Fig. 5 | Detection of MSI. a**, Summarized residual indel frequencies at mononucleotide microsatellites of NA12878. Arrows in the magnified box indicate overall fractions of reads with incorrect microsatellite lengths. **b**, Residual indel frequencies at mononucleotide microsatellites with different lengths from 8 to 18 nucleotides. **c**, Tumor and normal samples of a colon cancer patient with MSI were sequenced and diluted in silico to simulate MSI detection across different sample ratios. MSI score indicates a sum of probabilities of being an MSI site.

(anti-PD1/PDL1) but challenging to detect at low tumor fraction such as from liquid biopsies[40]. Indeed, CODEC WGS on NA12878 showed lower frequencies of both insertions and deletions at mononucleotide microsatellites than standard WGS (Fig. 5a). The fraction of CODEC reads with incorrect microsatellite lengths was 0.45%, which was 12 times lower than that of standard WGS. Such lower frequencies were consistent with what was observed across mononucleotide microsatellites of varied lengths from 8 to 18 nucleotides (Fig. 5b).

We then sequenced the MSI sample from the colon cancer patient and its matching normal sample to simulate MSI detection across different tumor fractions. When detecting MSI from an in silico dilution series, MSI score, which is a sum of probabilities of being an MSI site, of standard WGS paired with MSMuTect analysis[41] met its baseline at 0.1% (Fig. 5c). CODEC was at least ten times more sensitive than standard WGS with its MSI score higher than the baseline at 0.01%, implying that CODEC has potential for MSI detection at lower tumor fractions.

## Discussion

By physically linking both strands of each DNA duplex, CODEC transforms standard NGS instruments into massively parallel single duplex sequencers. We first showed that CODEC is as accurate as duplex sequencing but with a much lower sequencing requirement. This enabled more precise genetic testing with substantially fewer reads while uncovering biologically significant mutations from single DNA molecules, which are ordinarily obscured in NGS. Moreover, the applicability of CODEC across major NGS workflows ranging from targeted sequencing to WGS sets it apart from other high-accuracy NGS methods, which are confined to certain sites of the genome or shallow duplex depth.

One current limitation of CODEC is that long DNA fragments (for example, >300 bp) can be difficult to sequence with existing Illumina instruments for two reasons. First, CODEC links both strands together, so the total length of a CODEC library molecule is over twice as long, and current Illumina systems cannot effectively cluster long DNA molecules (>1 kb). Second, existing Illumina read lengths vary up to 300 bp per read, but each Watson and Crick strand sequence must be read to form a consensus of both strands.

Because CODEC retools standard NGS using a distinct molecular structure, there may still be room for improvement in sequencing

efficiency. For library construction, a new approach to attaching the adapter quadruplex instead of the standard NGS ligation chemistry used in this work could improve conversion into the correct structure. Applications involving hybridization capture could be further enhanced by achieving a higher on-target ratio. After one or two rounds of hybridization capture[42] on cfDNA (Supplementary Table 1), mean on-target ratios of CODEC were 28.8% and 71.2%, respectively (89.0% and 99.2% for duplex sequencing), which we believe can be improved by using a longer hybridization blocker with locked nucleic acid like commercial blockers for the CODEC linker. On the analysis side, a modified pipeline that treats byproducts as standard NGS data to supplement CODEC results may improve yields by reducing data loss.

CODEC could empower multiple facets of biomedical research and clinical diagnostic testing. One could apply CODEC to investigate rare biological events such as single-molecule mutations and mutational processes. CODEC could also reduce the cost of gene panel sequencing tests and enable broader swaths of the genome to be interrogated. Another promising area for CODEC is liquid biopsy testing, which often involves searching for low-abundance variants among large amounts of unmutated DNA.

As for whether CODEC could replace standard NGS for deep WGS, it is too early to conclude because there are no deep CODEC WGS data for a direct comparison. On one hand, CODEC WGS does not presently show higher sensitivity for high-abundance mutations that are ordinarily detected in deep (for example, 60×) standard WGS, but this was expected because the sensitivity for high-abundance mutations is mostly driven by NGS depth as opposed to accuracy. On the other hand, we have found high PPVs for CODEC single duplex reads, suggesting that deep CODEC may be able to uncover further true mutations, including those that are otherwise below the limit of detection of deep standard NGS. The challenge is that we currently lack a complete 'ground truth' for all mutations, making this difficult to evaluate. Further studies are thus needed to investigate whether deep CODEC WGS achieves higher overall sensitivity for rare mutations over a wider and deeper range of allele frequencies.

In all, CODEC opens new frontiers for NGS testing by enabling more precise genetic analysis at lower cost, and deeper biological insights to be gleaned from NGS data with single DNA molecule resolution. This will usher in a new era of DNA sequencing that is not only inexpensive

but also accurate, which is of great potential value for biomedical research and the broader use of clinical diagnostics.

## Online content

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

## Methods

### Ethical approval, DNA samples and oligonucleotides

All patients provided written informed consent to allow the collection of blood and/or tumor tissue and the analysis of clinical and genetic data for research purposes. The IRB of the Dana-Farber Cancer Institute and New York University Grossman School of Medicine approved these protocols. This research was conducted in accordance with the provisions of the Declaration of Helsinki and the U.S. Common Rule. Plasma from Dana-Farber Cancer Institute (DFCI) protocol 05–246 (breast cancer, patient 315)[23] and 13–383 (TBCRC 030) (breast cancer, patients 1–24) was derived from 20 cc whole blood in Streck tubes. Plasma from DFCI protocol 05–055 (breast cancer, patient 95)[23] was derived from 10 to 20 cc whole blood in EDTA tubes. Snap-frozen colon adenocarcinoma stage II/III and paired normal tissue biopsies from treatment-naïve patient (MSI, patient 19) were obtained from the Massachusetts General Hospital Tumor Bank, and gDNA was extracted using the Blood and Tissue kit (QIAGEN, 69556)[40]. The sperm sample was obtained as part of an IRB-approved human subjects protocol at New York University Grossman School of Medicine[43] (Supplementary Text). We have a limited amount of patient samples that may not be shared.

NA12878 was purchased from Coriell (NA12878). Fresh whole blood (10–20 cc) from appropriately consented healthy donors was obtained through Research Blood Components. All DNA samples were stored in low TE buffer (10 mM Tris–HCl, 0.1 mM EDTA, pH 8) and were fragmented by Covaris ultrasonicator to have a mean size of 150 bp except for cfDNA and ddBTP-blocked ER/AT. All CODEC adapter oligonucleotides were synthesized by Integrated DNA Technologies (IDT) and went through PAGE purification (Supplementary Table 5). The adapters for duplex sequencing were custom-ordered by the Broad Institute from IDT.

### End-repair/dA-tailing

Unless noted otherwise, we used NEBNext Ultra II DNA Library Prep Kit for Illumina (NEB, E7103L) for ER/AT. Duplex-Repair[32] consists of four steps. In step 1, DNA is treated with an enzyme cocktail consisting of EndoIV (NEB, M0304S), Fpg (NEB, M0240S), UDG (NEB, M0280S), T4 PDG (NEB, M0308S), EndoVIII (NEB, M0299S) and ExoVII (NEB, M0379S) in NEBuffer 2 in the presence of 0.05 µg µl⁻¹ BSA at 37 °C for 30 min. In step 2, T4 PNK (NEB, M0201S), T4 DNA polymerase (NEB, M0203S), 0.8 mM ATP and 0.5 mM dNTP mix are added into the step 1 reaction mix and incubated at 37 °C for another 30 min. In step 3, HiFi Taq ligase (NEB, M0647S) and 10× HiFi Taq ligase buffer are spiked into the step 2 reaction mix and incubated on a thermal cycler that heats from 35 °C to 65 °C over the course of 45 min. The resulting products are purified by performing 3× AMPure XP (Beckman Coulter, A63882) cleanup. In step 4, the purified products are treated with Klenow fragment (3′–5′ exo-) (NEB, M0212L) and Taq DNA polymerase (NEB, M0273S) in NEBuffer 2 in the presence of 0.2 mM dATP at room temperature for 30 min followed by 65 °C for 30 min. We modified ddBTP-blocked ER/AT[22] to start with fragmentation with HpyCH4V (NEB, R0620S) and AluI (NEB, R0137S; Extended Data Fig. 6b) in NEB CutSmart buffer at 37 °C for 15 min, purified with 2.5× AMPure XP beads. Fragmented DNA was dA-tailed in NEBuffer 4 with Klenow fragment and dATP/ddBTP (ddTTP, ddCTP and ddGTP) at 37 °C for 30 min.

### CODEC

The CODEC adapter quadruplex was prepared by diluting four 100 µM oligonucleotides to 5 µM with low TE buffer and 100 mM NaCl, followed by heating at 85 °C for 3 min, cooling with −1 °C per min to 20 °C, and incubating at room temperature for 12 h. Mastercycler X50 (Eppendorf) and Axygen 0.2 ml Maxyumum Recovery PCR tubes (Corning, PCR-02-L-C) were used for the annealing. The annealed adapter complex was kept at −20 °C for future use. Input DNA for CODEC ranged from 2.5 ng to 20 ng, although we observed the correct products with as little as 0.01 ng mass (Extended Data Fig. 10). We used NEBNext Ultra II DNA

Library Prep Kit for Illumina and followed the manufacturer's manual with several exceptions–(1) AMPure XP cleanup with 1.8× volume ratio was performed before the ligation step, followed by elution with 40 µl of low TE buffer (not included in CODEC v0 protocol; Supplementary Table 1); (2) ligation time was increased to 1 h, 5 µM adapter complex was diluted with adapter dilution buffer (10 mM Tris–HCl, 1 mM EDTA, 10 mM NaCl, pH 8) to 500 nM before use and replaced NEB adapter; (3) 3 µl of 5′-deadenylase (NEB, M0331S) were added to ligation reaction; (4) strand displacing extension (sample 40 µl, 10× buffer 10 µl, 0.2 mM dNTP, polymerase 1 µl, nuclease-free water up to 100 µl) was performed with phi29 DNA polymerase (NEB, M0269L) at 30 °C for 20 min, followed by 0.75× AMPure XP cleanup; (5) KAPA HiFi HotStart ReadyMix (Roche, 07958935001) and xGen Library Amplification Primer Mix (IDT, 1077677) were used for PCR by following the manual of KAPA Library Amplification Kit with 2 min of extension and (6) 0.65× AMPure XP cleanup was performed twice after the PCR. For cfDNA, the second cleanup followed the double-size selection protocol with 0.5× and 0.65× volume ratios.

Illumina libraries for standard NGS and duplex sequencing were prepared with KAPA HyperPrep Kit (Roche, 07962363001). All library preparations were performed on twin.tec PCR Plates LoBind 250 µl (Eppendorf, 0030129504). Library quantitation was performed with Qubit dsDNA HS kit (Invitrogen, Q33230) paired with Bioanalyzer DNA High Sensitivity chips (Agilent, 5067-4626).

### Enrichment

All enrichment was performed with xGen Hybridization and Wash kits (IDT, 1072281) and xGen Blocking Oligos (IDT, 1075476), following the manufacturer's manual except for the mass of input DNA, which was 1000 ng. Some samples went through two rounds of capture (Supplementary Table 1). xGen Pan-cancer Panel (IDT, 1056205) Custom WES panel was ordered by the Broad Institute from Twist Bioscience, and the other panels, including xGen Pan-cancer Panel (IDT, 1056205), were ordered from IDT.

### Sequencing

Standard NGS and duplex sequencing were performed with Illumina HiSeq X (2 × 151 cycles) and NovaSeq 6000 (2 × 100, 2 × 151, or 2 × 250 cycles). CODEC was performed with Illumina NovaSeq 6000 (2 × 166 or 2 × 260 cycles).

### Standard WGS tumor/normal data processing

We sequenced nine breast cancer tumors (fresh-frozen) with at least 60× coverage and matched germline control with 15× coverage. Somatic SNVs and indels were called by aligning FASTQ files to HG19 using BWA[44]. The aligned reads were then processed by PICARD MarkDuplicate. After deduplication, the base quality scores were recalibrated by GATK BQSR[45]. Finally, the small variants were called using Mutect2 (ref. 34) from cloud using Terra (https://app.terra.bio/#workspaces/help-gatk/Somatic-SNVs-Indels-GATK4). We used all default parameters and are thus referred to as standard+Mutect2 mutations in this study.

### CODEC data processing

Due to the unique CODEC read structure, we developed CODECsuite to process CODEC data (Supplementary Text). CODECsuite is written in C++14, R v4.1 and python v3.7, and we use Snakemake v7.3.8 (ref. 46) as the workflow management system. CODECsuite consists of three major steps: demultiplexing, adapter trimming and single-fragment mutation calling. The workflow also involves other standard tools such as BWA v0.7.17, Fgbio v2.0.2, GATK v4.3.0, Picard v2.27.1 and SAMtools v1.15.1 (ref. 47). Illumina bcl2fastq was used to generate FASTQ files (with -R -o, no --sample-sheet because CODECsuite will demultiplex) but was not included in the suite. After demultiplexing and adapter removal, we mapped the raw reads using BWA against human reference hg19. Fgbio was then used to collapse the PCR duplicates. These final consensus

reads were then mapped to the reference genome using BWA again. An indel realignment step using GATK was added to the workflow of capture data but was excluded from the WGS workflow. The entire workflow and more details are available on the CODECsuite GitHub.

## Duplex sequencing data processing

Fgbio was used to generate duplexes consensus and to filter the consensus reads[23,48]. Read families with at least two copies of each strand were required for generating duplex consensus except for duplex sequencing WGS, which relaxed the requirement to one copy of each strand to get the best possible duplex recovery.

## Residual SNV frequencies in capture sequencing

To identify residual SNVs, we mask all known germline and somatic mutations (for example, cancer mutations that are found in tumor sequencing and CH mutations confirmed by mutation validation). The duplex BAMs from both cfDNA and matched normal samples (from buffy coat) were generated in the same way and were applied to the same set of filters as follows: (1) no secondary and supplementary alignments; (2) Mapq ≥60; (3) Levenshtein distance (L-distance) between the reads excluding soft clipping and reference genome ≤5 and number of non-N-base L-distance ≤2 and (4) excluding bases within 12 bp distance from both fragment ends. To not confuse errors with real mutations, we precomputed the germline SNPs using GATK4 (HaplotypeCaller) from the duplex sequencing normal samples as they have a higher on-target ratio and hence coverage. For the patient sample, we found three somatic SNVs (median, VAF = 0.26; range, 0.24–0.28) in the captured regions (Supplementary Table 6) using Mutect2. Those somatic mutations (patient sample only) and germline mutations were masked when calculating the residual SNV frequencies. In addition, we used duplex sequencing buffy coat samples to filter CH mutations. We found two CH mutations in both patient's and healthy donor's buffy coat samples. All of the CH mutations are found in *KMT2C* genes. These sites were also excluded when calculating the residual SNV frequencies. Finally, the specificity checks were performed on cfDNA samples to remove substitutions that may arise from alignment errors[23,48].

## CODEC single-fragment mutation caller

We call mutations from each read family. One read family corresponds to one unique DNA molecule. Bioinformatically, a read family is defined as a collection of pair-end reads with the same start, stop mapping positions and UMI sequences (if UMI is used). The number of read pairs (R1 + R2 count as one read pair) in the family is called family size. The duplex sequencing protocol requires at least two read pairs from each strand to form duplex consensus and thereby family size ≥4. Whereas, for CODEC, a single read pair is sufficient to form a duplex and thus family size ≥1. In this study, we also applied the CODEC single-fragment mutation caller to standard WGS when compared to CODEC data to control the impact from the analytical pipeline. The standard WGS is usually deduplicated, which means that a single read pair is selected as the primary alignment to represent the whole family when family size >1. In this case, CODEC single-fragment caller was applied to primary alignments.

The advantage of CODEC over duplex sequencing is that we can call mutations with high accuracy (Q70) from families of size 1. To achieve this, we need to select good clusters from the flow cell because the error rates of Illumina vary from surface to surface, tile to tile and cluster to cluster[49]. Thus, we implemented a few fragment-level and base-level filters (Supplementary Text).

For all patients, we sequenced matched peripheral blood cells to 15× or greater depth to omit germline polymorphisms. When a germline bam is provided (in this study we used 10–15× WGS), we require (1) the SNV and indel should not be found in the germline bam. Because one indel can have many equivalent alignments, we require that the interrogated indel is not found in a small window (default, 10 bp) of the germline bam. (2) At least five unique fragments cover the SNV or indel site in the germline bam. All of these thresholds are tunable parameters in the analysis pipeline. Using tighter thresholds can increase the precision but will also decrease the sensitivity. This allows users to customize based on their particular application. If no germline bam is available, the user should provide a germline VCF to mask the germline mutations. In this paper, we also used the gnomAD[50] VCF to mask all common germline SNPs and indels (AF > 0.01%) to mitigate possible contamination[22]. When evaluating the CODEC NA12878 WGS, we used v3.3.2 GIAB NA12878 high confidence VCF and BED file as germline masks and evaluation regions, and no matched germline was used. For all other WGS analyses except for MSI calling, we used the GIAB V3.0 easy regions (total 2.3B bases) to call mutations.

## Germline SNV and small indel calling in downsampled WGS

We first downsampled CODEC and standard WGS NA12878 samples to 1–5× (step size 1×) mean coverages using GATK DownsampleSam. Then for CODEC data, we first removed all byproducts. Next, we ran GATK4.1.4.1 best practices pipeline via Cromwell and Terra workflow (available at web resources and computed on the Google Cloud Platform). Note that the actual coverages used by GATK were thus lower for CODEC (0.6–3.0×) versus standard Illumina NGS. We used RTG vcfeval[51] to calculate FP and FN for SNPs and indels (<50 bp) without penalizing genotyping errors (if heterozygous variants were called as homozygous and vice versa) using v3.3.2 high confidence VCF and BED file as input. We then calculated FP per million bases by normalizing against the high confidence region size and FN ratio by dividing FN by the total number of true variants.

## Mutation validation

CODEC single-fragment mutation caller was used to call somatic SNV (SSNV) mutations from five breast cancer patients' tumor WGS samples, eight breast cancer patients' buffy coat WGS samples, four breast cancer patients' cfDNA WGS samples and four healthy donors' cfDNA WGS samples. Our in-house probe design pipeline[23] was used to design probe sequences for hybrid capture of CODEC whole-genome SSNV. We created three sets of hybridization capture data for validation as follows: (1) CODEC SSNVs using duplex sequencing libraries constructed from the same samples that the SSNVs were derived, (2) SSNVs called from CODEC WGS on cfDNA in CODEC and duplex sequencing cfDNA libraries and (3) the same SSNVs as in (2) but starting with 500 ng of buffy coat DNA for library construction. Designed probes were purchased from IDT and hybrid capture was performed with a 25× multiplicity DNA input of 6 µg. The numbers of probes/SSNVs in each case are included in Supplementary Table 1.

## Mutation signatures analysis

The single-fragment mutations and highly abundant mutations called from Mutect2 were used as input for deconstructSigs[52] to calculate residual mutation frequencies in the 96 TNC and to derive mutation signatures. In brief, deconstructSigs is a reference-based approach, which finds a set of reference signatures $S$ (for example, COSMIC signatures) that minimizes the sum-squared error between the reconstructed spectrum $WS$ and the input spectrum $T$. deconstructSigs calls $W$ the weight matrix. In other literature, $W$ is more commonly called an exposure matrix. Because we are limited by the number of samples (sometimes only one sample), this reference-based approach is more appropriate than a de novo approach like SignatureAnalyzer[53] or SigProfiler[54]. For MSI signatures, we used COSMIC V3.2 as the reference panel. While for the HRD signature (signature 3 in COSMIC V2 or SBS3 in COSMIC V3), we used COSMIC V2 as the reference panel as people have found signature 3 in COSMIC V2 resembles the HRD signature generated in vivo more than the SBS3 in COSMIC V3 does[55]. To further improve signature 3 detection in the breast cancer patient samples, we restricted to only breast-cancer-related signatures defined by HRDetect[56]. CHORD[39]

results on full coverage standard Illumina NGS were used to indicate the HRD status of the eight breast cancer patients. The plots of the 96-TNC spectrum were generated by sigfit[57].

### MSI detection

The full-coverage CODEC consensus BAM and full-coverage standard WGS R1R2 consensus BAM on NA12878 were compared against each other to demonstrate CODEC ability to correct PCR stutter errors and thus to reduce background noise for MSI detection. MSIsensor-pro[58] was used to scan the hg19 for homopolymers of size 8–18 nt. Because MSIsensor-pro does not have mapping quality or secondary alignments filters, we prefiltered the BAM using SAMtools by requiring mapq ≥60 and no secondary or supplementary alignments. And then MSIsensor-pro was used again to count the number of reads that support different lengths of homopolymer at those preselected sites. We removed any homopolymer sites that overlap or are in proximity (±5 bp) with any germline variants. After that, the reference lengths of the homopolymer sites were considered true lengths, and observed length distributions from reads were compared against them.

We resequenced an MSI-H tumor sample and the matched adjacent normal sample from a colorectal cancer patient using both CODEC (5×) and standard Illumina NGS (12×). Previous study[40] has found the tumor purity to be 66%. We, therefore, in silico mixed the reads from MSI samples and reads from the normal samples for CODEC and standard NGS separately to mimic a dilution series of tumor purities from 66% to 0.01% at a depth of 2× at each dilution point. We developed an MSI caller called CODEC-MSI, which uses a Bayesian Genotype Quality model to calculate a posterior probability of being MSI site for each locus of interest (Supplementary Text). Because it uses a tumor-normal pair, we split the normal samples into two halves and no reads were shared between the two splits. One of the splits was used for the mixing and the other was used as the normal input of the tumor-normal pair for CODEC-MSI. The pure normal samples were indicated as 0% in Fig. 5c. The MSI scores of the standard WGS were calculated by MSMutect[41], which are also a sum of per-site MSI scores.

### Statistics and reproducibility

The sample size was determined by the availability of tissue and the cost of the experiment. No statistical method was used to predetermine the sample size. The experiments were not randomized. The investigators were not blinded to allocation during experiments and outcome assessment. All statistical analyses in this work can be reproduced by codes on our GitHub repository (https://doi.org/10.5281/zenodo.7705860).

### Reporting summary

Further information on research design is available in the Nature Portfolio Reporting Summary linked to this article.

## Data availability

DNA sequencing data and results generated for this study such as Mutect2 MAF files will be available from dbGaP under accession code phs003255.v1.p1. NA12878 PacBio data was downloaded from GIAB https://ftp-trace.ncbi.nlm.nih.gov/ReferenceSamples/giab/data/NA12878/PacBio_SequelII_CCS_11kb/.

## Code availability

Code required to reproduce the analyses in this paper is available online. CODECsuite is available at https://doi.org/10.5281/zenodo.7705860 and https://github.com/broadinstitute/CODECsuite, which also contains the end-to-end Snakemake workflow and code for CODEC-MSI. Other software used includes bcl2fastq (v2.20, https://support.illumina.com/downloads/bcl2fastq-conversion-software-v2-20.html); BWA (v0.7.17); SAMtools (1.15.1); GATK (4.3.0, https://github.com/broadinstitute/gatk); FGBIO (v2.0.2, https://github.com/fulcrumgenomics/fgbio); Picard (v2.27.1, http://broadinstitute.github.

io/picard/); Snakemake (v7.3.8); GATK HaplotypeCaller germline calling best-practice (v4.1.4.1, https://app.terra.bio/#workspaces/warp-pipelines/Whole-Genome-Analysis-Pipeline); GATK somatic calling best-practice (4.1.7.0, https://gatk.broadinstitute.org/hc/en-us/articles/360035894731-Somatic-short-variant-discovery-SNVs-Indels-); Sigfit (v2.2); DeconstructSigs (v1.9.0); R (v4.1); Python (v3.7); ggplot2 (v3.3.5); tidyverse (v1.3.1); data.table (v1.14.2); Pandas (v1.3.3); Pysam (v0.16); seaborn (v0.11.2). HRDetect's breast cancer signature set is available at https://github.com/Nik-Zainal-Group/signature.tools.lib#examplese01.

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

## Acknowledgements

The authors acknowledge the Gerstner Family Foundation for its generous support. We are grateful for the funding support to the TBCRC from The Breast Cancer Research Foundation and

Susan G. Komen. This study was also supported in part by SPARC award from the Broad Institute. G.D.E. and J.E.S. receive support from NIH grant R21HD105910. G.D.E. receives support from NIH grant DP5OD028158, the Sontag Foundation and the Pew Foundation. M.G-P. receives support from NIH training grant T32AG052909 and NIH grant NIA F32AG076287. J.E.S. receives support from Bristol Myers Squibb Foundation, Damon Runyon cancer research foundation and Vinney scholars award. S.T. receives support from the ASCO Conquer Cancer Foundation Young Investigator Award (2021YIA-5688173400) and the Prostate Cancer Foundation Young Investigator Award (2021). S.P. receives funding through T32 (T32HL116275). G.M.M. receives support from NIH grant R01 CA221874. The funders had no role in study design, data collection and analysis, decision to publish or preparation of the manuscript. We thank Erin LaRoche for the helpful discussion on NGS.

## Author contributions

V.A.A. and J.H.B. conceived and designed the project. V.A.A., J.H.B., and R.L. developed the method. G.D.E., M.G-P., E.L., M.H.L., G.M.M., E.L.M., H.A.P., J.E.S. and A-B.S. collected DNA samples. J.H.B., E.N. and E.R. performed experiments. R.L. analyzed the data. V.A.A., Z.A., J.H.B., T.B., J.C., G.D.E., G.G., R.L., E.N., S.P., J.R., E.R., D.S., J.E.S., S.S., S.T. and K.X. interpreted the data. V.A.A., J.H.B., T.B., G.D.E., T.R.G., R.L., G.M.M., E.R., J.R., D.S., J.E.S., S.T. and K.X. wrote and reviewed the manuscript.

## Competing interests

V.A.A., J.H.B., R.L. and G.M.M. have filed a patent application (PCT/US2021/062966) on this method. T.R.G. has paid scientific advisory roles and equity in Dewpoint Therapeutics and Anji Pharmaceuticals, holds founder's equity in Sherlock Biosciences, is a paid advisor to Braidwell Inc. and has research funding from Bayer HealthCare, Calico Life Sciences and Novo Holdings. J.E.S. is the key opinion leader for ForTec Medical. The remaining authors declare no competing interests.

## Additional information

**Extended data** is available for this paper at https://doi.org/10.1038/s41588-023-01376-0.

**Correspondence and requests for materials** should be addressed to Viktor A. Adalsteinsson.

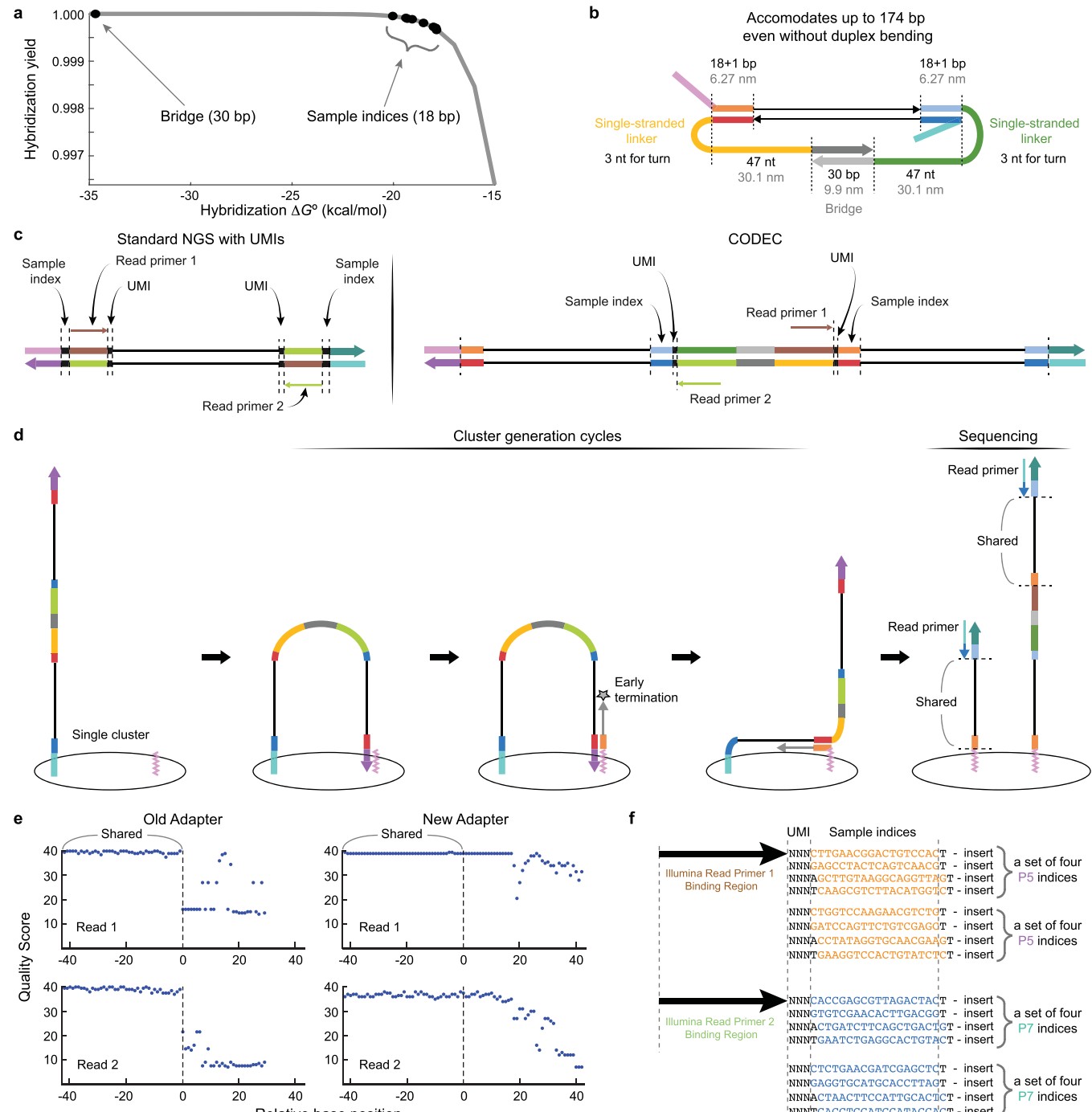

**Extended Data Fig. 1 | Design principle of CODEC adapter quadruplex.**
(**a**) Predicted hybridization yield of the double-stranded regions with oligonucleotide concentrations of 500 nM at 20 °C and [Na⁺] = 10 mM. (**b**) The length of single-stranded linkers was determined to mitigate bending stiffness of a target duplex. Duplexes with up to 174 bp can be accommodated without bending at all, which was calculated using the lengths of DNA in B-DNA helix and single-stranded structure. Approximately, it is 0.33 nm per base pair along the helical axis of B-DNA and 0.64 nm per nucleotide for single-stranded DNA. We excluded 3 nucleotides from each single-stranded region, which is the minimum length of a hairpin loop. (**c**) Read primer binding sites of standard NGS and CODEC. (**d**) During Illumina cluster generation cycles, early termination in the middle of the insert region could create byproducts which turn into shorter fragments with only one insert. If a read primer binding site is located

at the end of a fragment, unlike CODEC, these subclonal fragments cause mixed fluorescence after sequencing cycles pass the shared region, and consequently, low Quality Scores. (**e**) Mean Quality Scores of each sequencing cycle by taking 42 bp before and after the shared region from random 100 read pairs. Before redesigning the adapter structure, Quality Scores suddenly dropped after the shared region. This issue was solved by moving the read primer binding regions to the linker to 'silence' all byproducts without the linker. (**f**) UMIs and each set of four sample indices are designed to collectively include all four bases at each base position while keeping similar hybridization $\Delta G°$ for high-quality image analysis of Illumina sequencers. For example, Illumina software uses up to first 25 bp for various purposes such as cluster identification, phasing correction, and chastity filter.

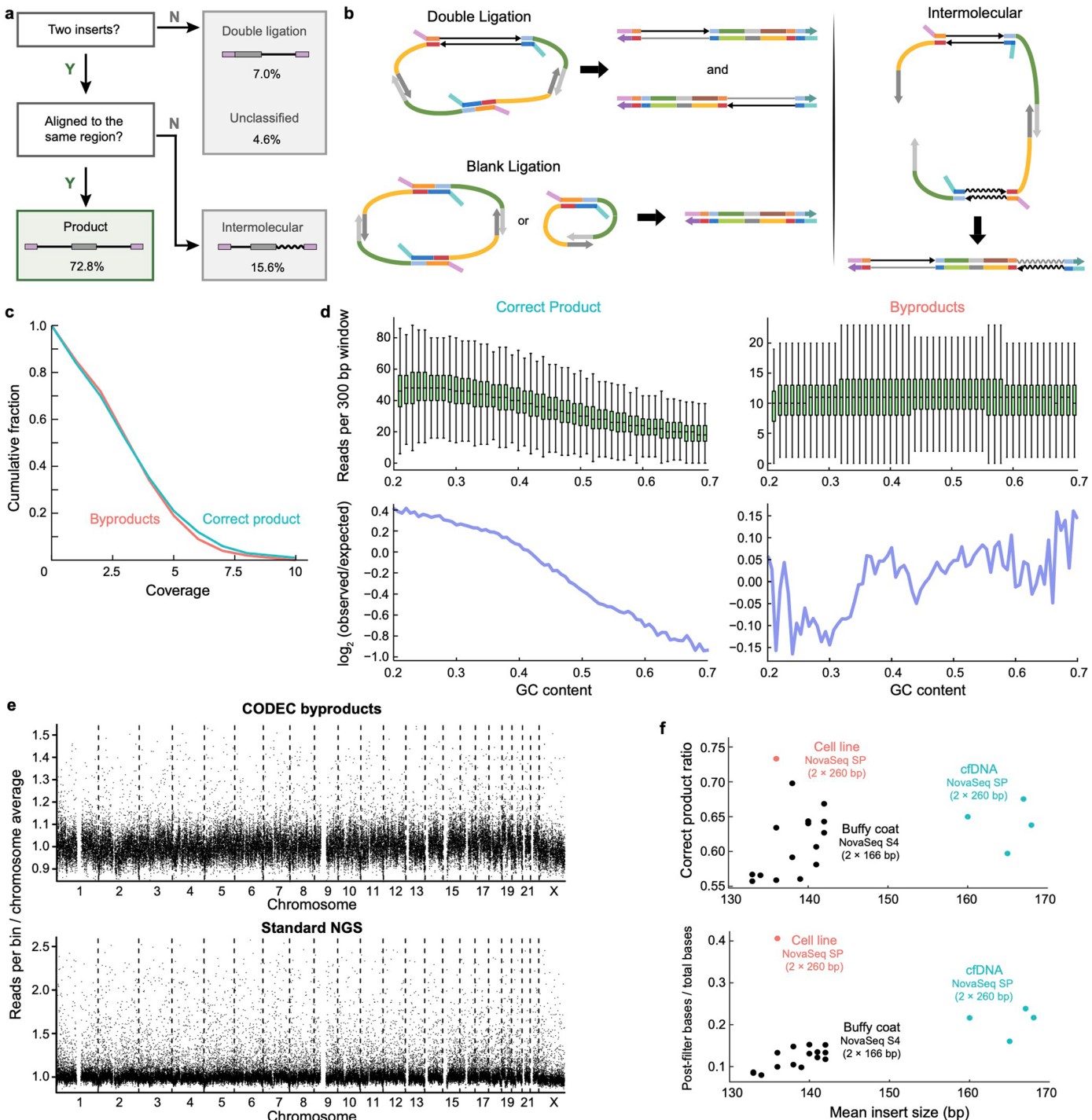

**Extended Data Fig. 2 | Byproduct analysis. (a)** Ratios of the correct CODEC product and byproducts which have been named after how they were likely created. **(b)** Expected mechanisms of byproduct formation. 'Double ligation' can occur when two adapter complexes are ligated to each end of an insert and go through T/T mismatched ligation with each other, as opposed to A/T ligation. 'Blank ligation' can occur when one or two adapter complexes go through T/T mismatched ligation with no insert. 'Intermolecular' can occur when polymerase extension uses another ligation product as a template instead of the opposite strand. **(c)** Cumulative fraction of sequencing coverage based on byproducts and the correct product reads. Their similarity implies that byproducts were randomly generated. **(d)** Medians of reads allocated to 300 bp windows grouped by their GC contents (top row) and their observed/expected

ratios (bottom row). Shorter lengths of byproducts may have mitigated GC bias of polymerases. Center lines, boxes, and whiskers indicate medians, 25% and 75% percentiles, and 5% and 95% percentiles, respectively. **(e)** The ratio of GC-corrected read counts per 50 kb bin, normalized by the LOESS-fitted (by GC) chromosome-wide mean value. CODEC byproducts and standard NGS reads from NA12878 gDNA were analyzed by ichorCNA. CODEC byproducts showed lower normalized values than standard NGS, suggesting that there were no particular genomic regions with higher fractions of byproducts. **(f)** Correct product ratio and percentage of bases that passed all analysis filters vs. mean insert size of each library. Bases in byproducts were counted towards total bases, but not towards post-filtered bases.

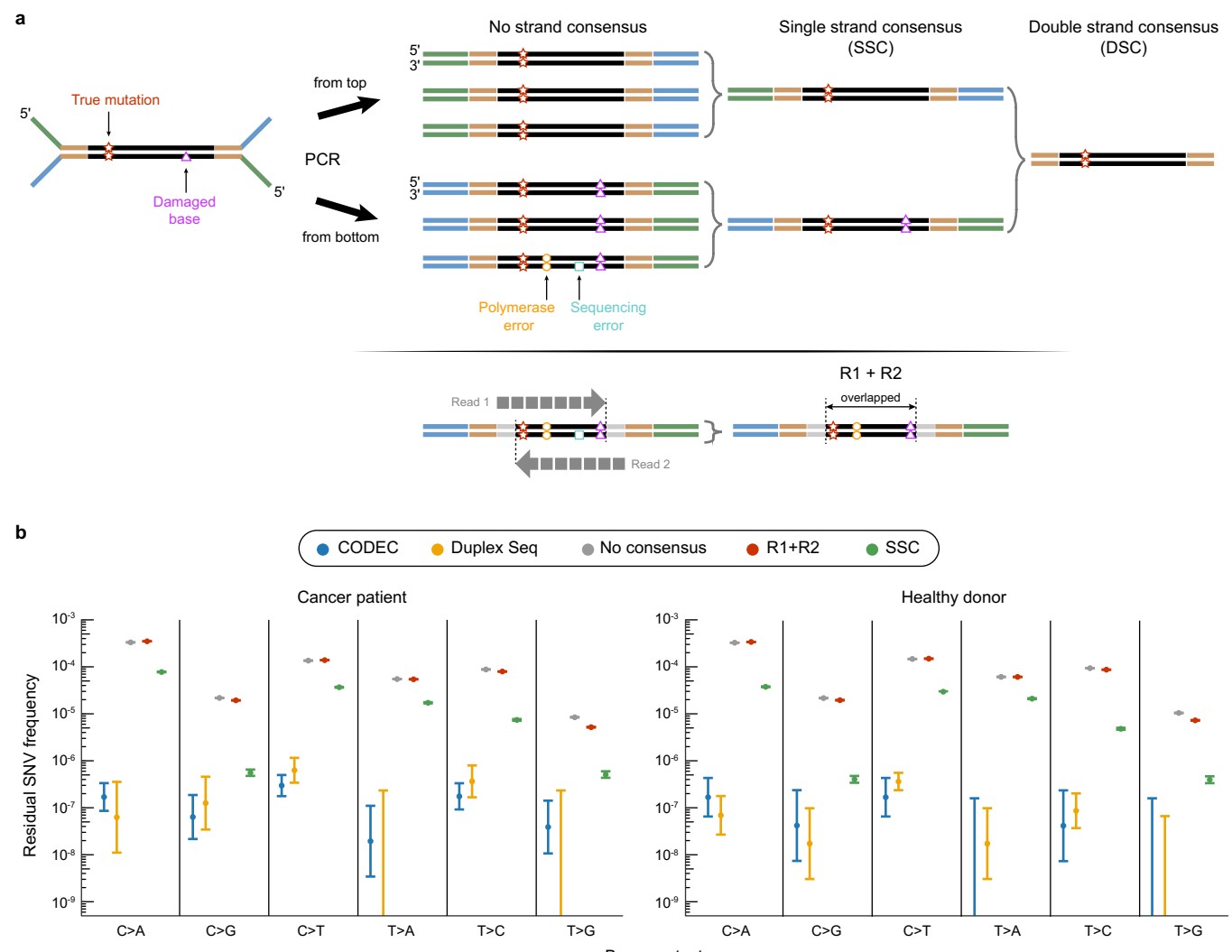

**Extended Data Fig. 3 | Different types of strand consensuses. (a)** 'No strand consensus' treats each read of a read-pair as an independent read. 'Single strand consensus' is generated by collapsing multiple reads from the same strand of an original molecule, which cannot distinguish damaged bases from true mutations. 'R1 + R2' is a consensus between read 1 and 2 of paired-end sequencing, which both read the same library molecule from one strand of an original molecule. It does not suppress errors other than sequencing errors. **(b)** Residual SNV frequency per base context of targeted deep sequencing with the pan-cancer panel. Data points and error bars indicate mean values and 95% binomial confidence intervals by Wilson method, respectively.

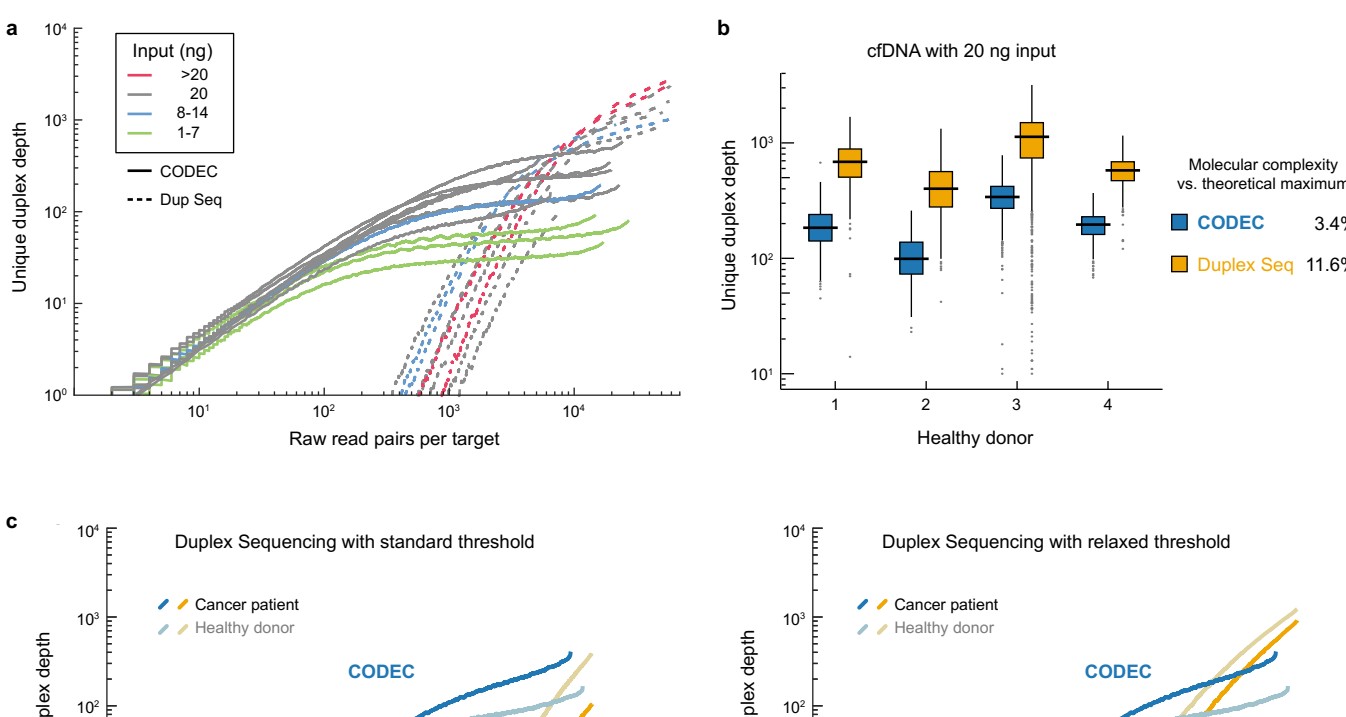

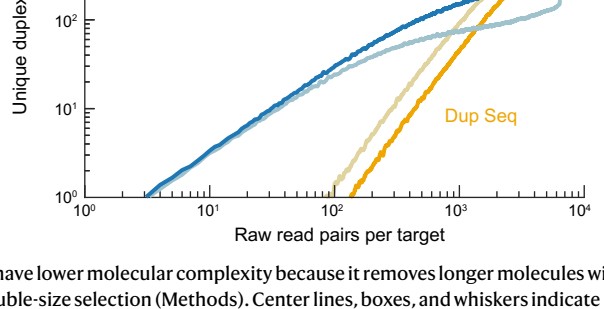

**Extended Data Fig. 4 | Unique duplex recovery by CODEC and duplex sequencing.** (**a**) Mean unique duplex depth vs. raw read pairs per target after performing hybridization capture with personalized probe panels on cfDNA libraries of breast cancer patients and healthy donors. Samples were grouped by their mass into library construction. (**b**) Mean unique duplex depths of cfDNA from four healthy donors which had the same input mass. We assumed that 20 ng input had 6000 haploid copies. When a sample is cfDNA, CODEC was expected

to have lower molecular complexity because it removes longer molecules with double-size selection (Methods). Center lines, boxes, and whiskers indicate medians, 25% and 75% percentiles, and 5% and 95% percentiles, respectively. (**c**) The effect of relaxing the threshold of duplex sequencing from two reads of each strand to one read of each strand, indirectly observed with the same data as Fig. 2a, b. Schmitt et al.[20]. and Abascal et al.[22]. required three and two reads of each strand, respectively.

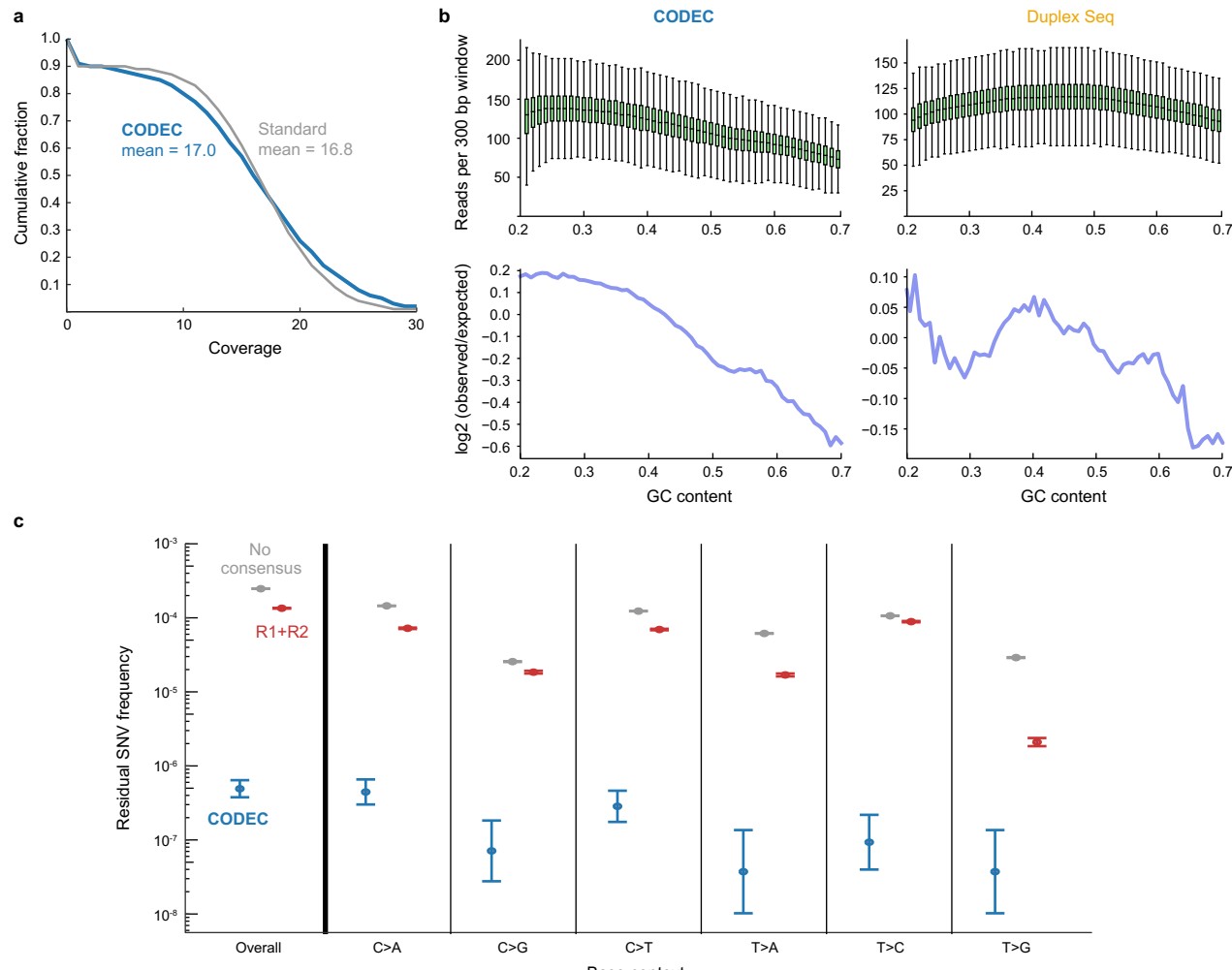

**Extended Data Fig. 5 | Details of CODEC WGS and WES. (a)** Cumulative fraction of sequencing coverage of CODEC and standard WGS with matching median coverage. The curves implied that the uniformity of CODEC WGS was not as high as that of standard WGS. **(b)** Medians of reads allocated to 300 bp windows grouped by their GC contents (top row) and their observed/expected ratios (bottom row). CODEC may have been affected more by polymerase's GC bias due to its longer fragment length. Center lines, boxes, and whiskers indicate medians, 25% and 75% percentiles, and 5% and 95% percentiles, respectively. **(c)** Overall residual SNV frequencies and their base contexts of WES on human gDNA sample. Data points and error bars indicate mean values and 95% binomial confidence intervals by Wilson method, respectively.

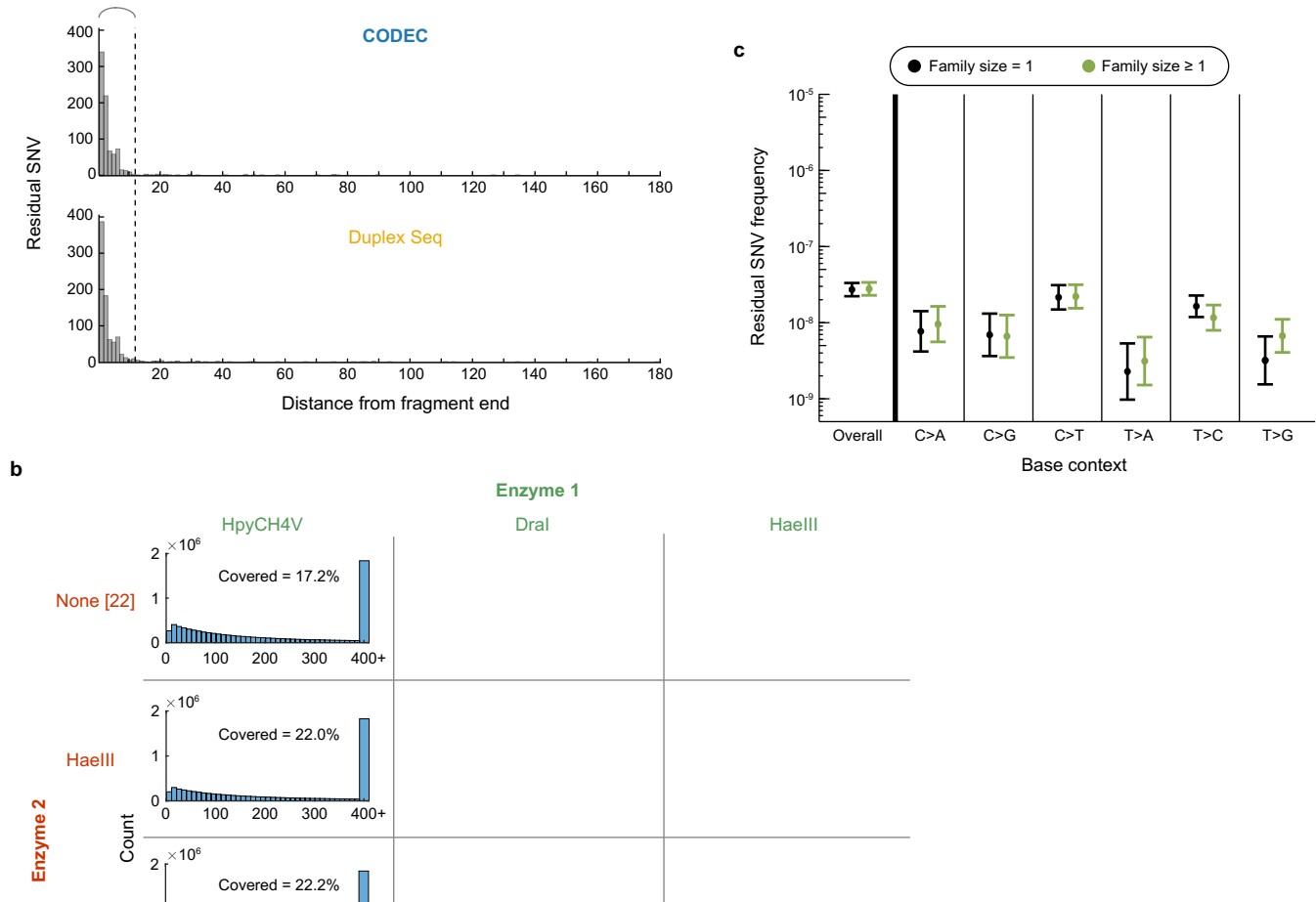

**Extended Data Fig. 6 | Details of suppressing errors at the end of DNA fragments before CODEC. (a)** Residual SNVs and their distances from fragment ends. This examples shows NGS data of a healthy donor after hybridization capture with the pan-cancer panel. We discard mutations within the last 12 bp from either end. **(b)** Theoretical fragment size distributions after double digestion with blunting restriction enzymes. Covered percentages show how much of human genome will turn into fragments with the size between 100 and 400 bp. The combination of HpyCH4V and AluI was selected for ddBTP-blocked ER/AT for Fig. 2e. **(c)** Residual SNV frequencies of CODEC paired with ddBTP-blocked ER/AT. Only using reads with a family size of one resulted in statistically the same SNV frequencies, confirming that a single read-pair is equally accurate. Data points and error bars indicate mean values and 95% binomial confidence intervals by Wilson method, respectively.

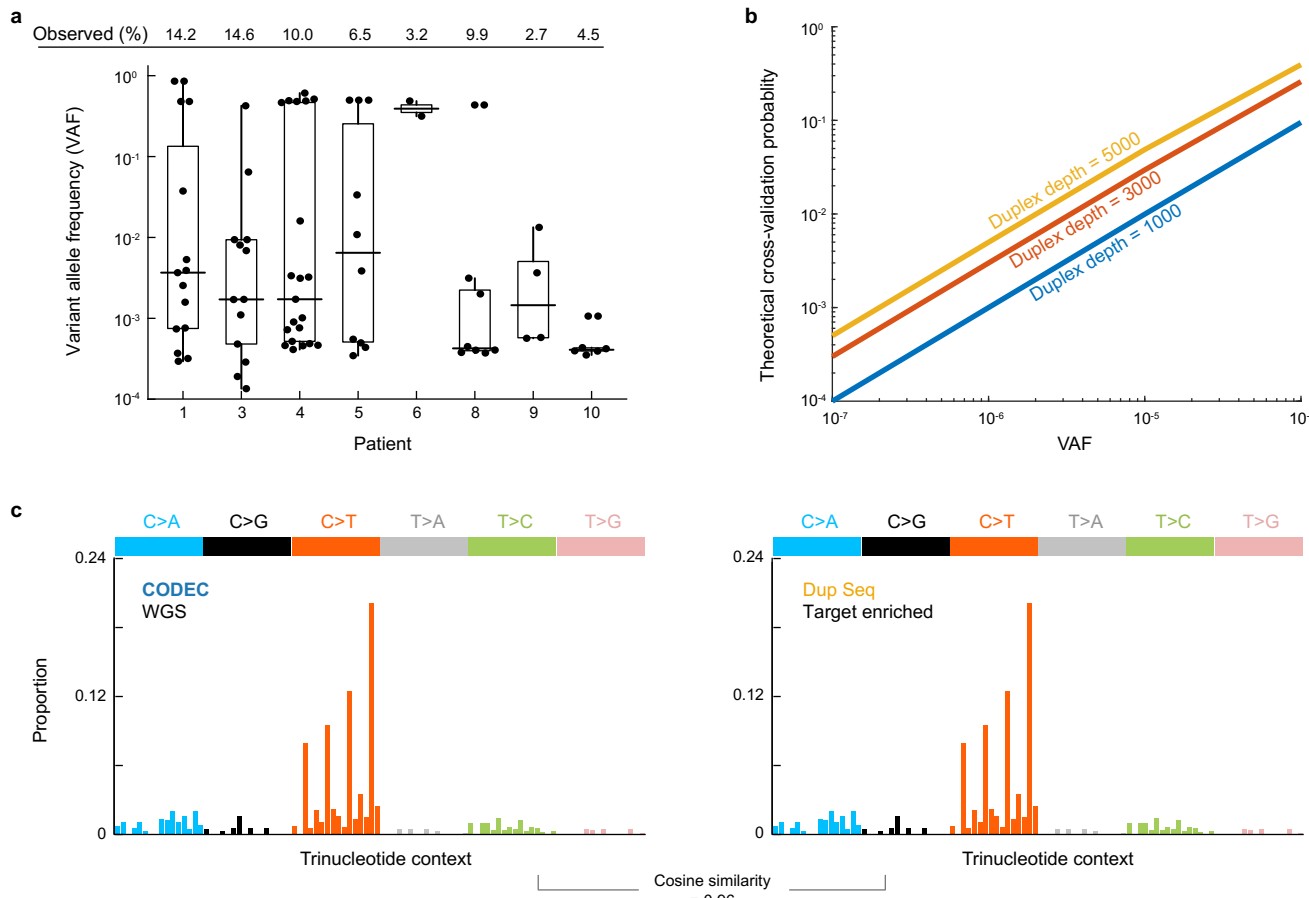

**Extended Data Fig. 7 | Cross-validation of the single-fragment mutations and their trinucleotide contexts. (a)** VAF of somatic mutations initially found in CODEC WGS on buffy coat DNA. The mutations were cross-validated by targeted deep sequencing using newly created duplex sequencing libraries from the same samples. Observed ratios show how much of mutations were observed again from the independent libraries. Center lines, boxes, and whiskers indicate medians, 25% and 75% percentiles, and 5% and 95% percentiles, respectively. **(b)** Theoretical probability of the cross-validation based on the binomial distribution. Because sampling a rare mutation in a biological sample is stochastic, somatic mutations with lower VAF are less likely to be validated. Considering most somatic mutations in buffy coat DNA aren't under positive selection pressure and have low VAF, only a subset of mutations identified by CODEC WGS will be sampled again for the independent libraries. **(c)** To investigate whether CODEC contributed to any new errors, we analyzed the trinucleotide error contexts of CODEC and duplex sequencing libraries from the same individuals in Fig. 3d. For duplex sequencing, hybridization capture data were used to acquire enough mutations for the analysis. After excluding mutations detected by Mutect2 or that has ≥2 duplex reads, CODEC and duplex sequencing had 3,992 and 204 mutations, respectively. One sample with high subclonality was also removed. The four highest peaks in each figure were cytosines in CpG contexts. The proportions of SBS1, which reflects spontaneous deamination of cytosines, were 24.2% and 19.7% for CODEC and duplex sequencing, respectively.

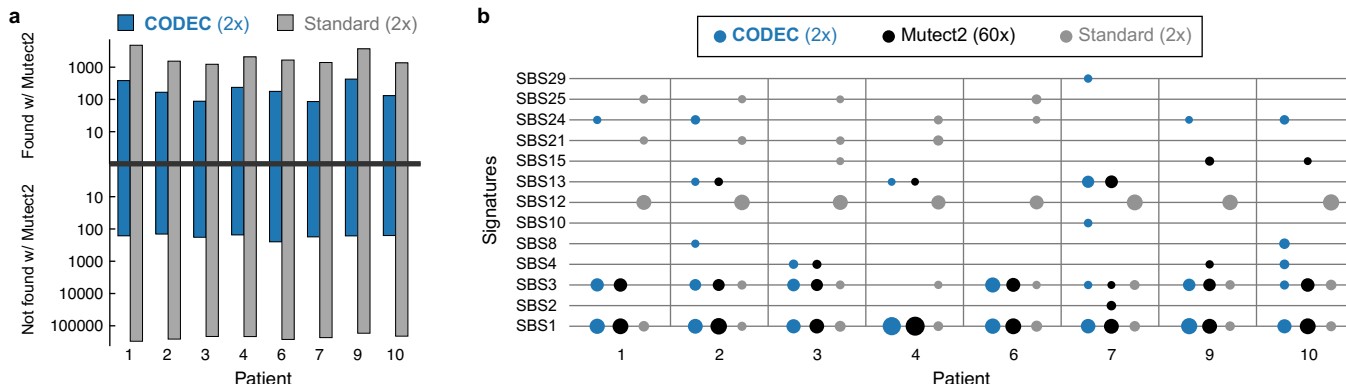

**Extended Data Fig. 8 | Mutations detected in breast tumor samples. (a)** Mutations detected by CODEC and standard WGS at 2× on eight breast tumor samples were validated by 60× standard WGS + Mutect2. **(b)** Full COSMIC signatures of eight breast tumor samples.

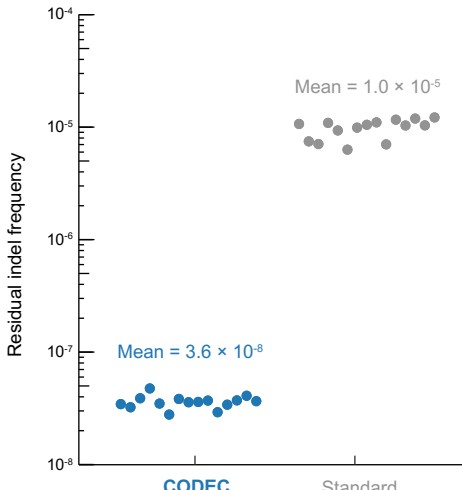

**Extended Data Fig. 9 | Residual indel frequency.** Residual indel frequencies of CODEC and standard WGS on buffy coat DNA of 15 breast cancer patients.

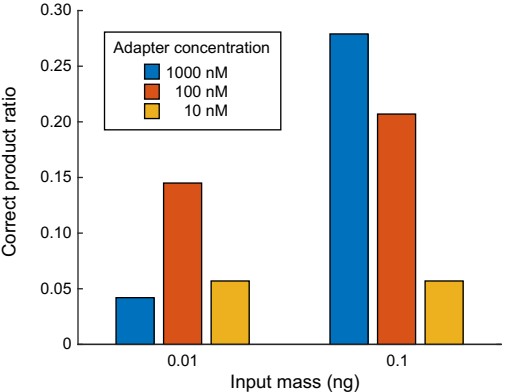

**Extended Data Fig. 10 | CODEC with lower input masses.** Three different concentrations of adapter were tested for 0.1 and 0.01 ng input mass. We used a mixture of 67, 100, 167 bp synthetic double-stranded DNA as insert, which resulted in distinct sizes between the correct product and byproducts. To estimate the ratios of the correct product, we measured the concentration of each peak with Bioanalyzer 2100 and High Sensitivity DNA Chip.

|---|---|

# Reporting Summary

## Statistics

For all statistical analyses, confirm that the following items are present in the figure legend, table legend, main text, or Methods section.

| n/a | Confirmed | |
|---|---|---|
| ☐ | ☒ | The exact sample size (*n*) for each experimental group/condition, given as a discrete number and unit of measurement |
| ☐ | ☒ | A statement on whether measurements were taken from distinct samples or whether the same sample was measured repeatedly |
| ☐ | ☒ | The statistical test(s) used AND whether they are one- or two-sided <br> *Only common tests should be described solely by name; describe more complex techniques in the Methods section.* |
| ☐ | ☒ | A description of all covariates tested |
| ☐ | ☒ | A description of any assumptions or corrections, such as tests of normality and adjustment for multiple comparisons |
| ☐ | ☒ | A full description of the statistical parameters including central tendency (e.g. means) or other basic estimates (e.g. regression coefficient) AND variation (e.g. standard deviation) or associated estimates of uncertainty (e.g. confidence intervals) |
| ☐ | ☒ | For null hypothesis testing, the test statistic (e.g. *F*, *t*, *r*) with confidence intervals, effect sizes, degrees of freedom and *P* value noted <br> *Give P values as exact values whenever suitable.* |
| ☐ | ☒ | For Bayesian analysis, information on the choice of priors and Markov chain Monte Carlo settings |
| ☒ | ☐ | For hierarchical and complex designs, identification of the appropriate level for tests and full reporting of outcomes |
| ☐ | ☒ | Estimates of effect sizes (e.g. Cohen's *d*, Pearson's *r*), indicating how they were calculated |

*Our web collection on statistics for biologists contains articles on many of the points above.*

## Software and code

Policy information about availability of computer code

| Data collection | No software was used. |
|---|---|
| Data analysis | CODEC data processing: https://doi.org/10.5281/zenodo.7705860 <br> bcl2fastq v2.2 <br> BWA v0.7.17 <br> Samtools v1.15.1 <br> Picard v2.27.1: http://broadinstitute.github.io/picard/ <br> fgbio v2.0.2: https://github.com/fulcrumgenomics/fgbio <br> GATK for germline: 4.1.4.1 <br> GATK/mutect2 for somatic: 4.1.7.0 <br> DeconstructSigs v1.9.0: https://github.com/raerose0l/deconstructSigs <br> Breast cancer signatures: https://github.com/Nik-Zainal-Group/signature.tools.lib#examplese01 <br> Sigfit v2.2: https://github.com/kgori/sigfit <br> Pipeline or workflow management: Snakemake 7.3.8 <br> Plot and data wrangling and statistic analysis: R (v4.1); Python (v3.7); ggplot2 (v3.3.5); tidyverse (v1.3.1); data.table (v1.14.2); Pandas (v1.3.3); Pysam (v0.16); seaborn (v0.11.2) |

For manuscripts utilizing custom algorithms or software that are central to the research but not yet described in published literature, software must be made available to editors and reviewers. We strongly encourage code deposition in a community repository (e.g. GitHub). See the Nature Portfolio guidelines for submitting code & software for further information.

## Data

Policy information about availability of data

All manuscripts must include a data availability statement. This statement should provide the following information, where applicable:

- Accession codes, unique identifiers, or web links for publicly available datasets
- A description of any restrictions on data availability
- For clinical datasets or third party data, please ensure that the statement adheres to our policy

Depositing DNA sequencing data and results generated for this study such as Mutect2 MAF files to dbGaP is ongoing. NA12878 PacBio data was downloaded from GIAB https://ftp-trace.ncbi.nlm.nih.gov/ReferenceSamples/giab/data/NA12878/PacBio_SequelII_CCS_11kb/.

# Field-specific reporting

Please select the one below that is the best fit for your research. If you are not sure, read the appropriate sections before making your selection.

☒ Life sciences ☐ Behavioural & social sciences ☐ Ecological, evolutionary & environmental sciences

For a reference copy of the document with all sections, see nature.com/documents/nr-reporting-summary-flat.pdf

# Life sciences study design

All studies must disclose on these points even when the disclosure is negative.

| | |
|---|---|
| Sample size | All clinical samples were selected based on their availability for this study (>20 ng for buffy coat, 1> ng for tumor DNA, and > 2.5 ng for cell-free DNA). |
| Data exclusions | In Extended Data Figure 7, a tumor sample with high subclonality was removed. |
| Replication | All experiments were performed independently and have no replicates. |
| Randomization | There was no allocating into experimental groups. |
| Blinding | There was no allocating into experimental groups. |

# Reporting for specific materials, systems and methods

We require information from authors about some types of materials, experimental systems and methods used in many studies. Here, indicate whether each material, system or method listed is relevant to your study. If you are not sure if a list item applies to your research, read the appropriate section before selecting a response.

## Materials & experimental systems

| n/a | Involved in the study |
|---|---|
| ☒ | ☐ Antibodies |
| ☐ | ☒ Eukaryotic cell lines |
| ☒ | ☐ Palaeontology and archaeology |
| ☒ | ☐ Animals and other organisms |
| ☐ | ☒ Human research participants |
| ☒ | ☐ Clinical data |
| ☒ | ☐ Dual use research of concern |

## Methods

| n/a | Involved in the study |
|---|---|
| ☒ | ☐ ChIP-seq |
| ☒ | ☐ Flow cytometry |
| ☒ | ☐ MRI-based neuroimaging |

## Eukaryotic cell lines

Policy information about cell lines

| | |
|---|---|
| Cell line source(s) | NA12878 (purified DNA purchased from Coriell) |
| Authentication | Coriell uses multiplexed PCR for 6 autosomal microsatellite markers. |
| Mycoplasma contamination | Cultures from Coriell are found free of mycoplasma. |
| Commonly misidentified lines (See ICLAC register) | N/A |

# Human research participants

Policy information about studies involving human research participants

| | |
|---|---|
| Population characteristics | Patients from Dana-Farber Cancer Institute protocols 05-246, 13-383, and 05-055 had breast cancer. A pair of colon cancer and adjacent normal colon samples were obtained from the Massachusetts General Hospital Tissue Bank. Healthy donor plasma and whole blood was obtained from Research Blood Components. Sperm sample of a 39-year-old donor was obtained from Cryos International. To our knowledge, donors were unrelated and samples were deidentified before being sent to us. |
| Recruitment | Patients were recruited from the Dana-Farber Cancer Institute in Boston, Massachusetts. Healthy donor biological material was obtained from Research Blood Components. Sperm sample was obtained from Cryos International. |
| Ethics oversight | The IRB of the Dana-Farber Cancer Institute and New York University Grossman School of Medicine approved these protocols. All patients provided written informed consent to allow the collection of blood and/or tumor tissue and analysis of clinical and genetic data for research purposes. |

Note that full information on the approval of the study protocol must also be provided in the manuscript.

