## [Peer Review File · Nature Genetics]

Peer Review Information

Manuscript Title: Single duplex DNA sequencing with CODEC detects mutations with high sensitivity

Corresponding author name(s): Dr Viktor Adalsteinsson

Reviewer Comments & Decisions:

Decision Letter, initial version:

17th May 2022

Dear Viktor,

Firstly, our sincere thanks to you and your co-authors for your patience during this prolonged process of peer review.

Your Technical Report, "CODEC enables 'single duplex' sequencing" has now been seen by 4 referees (please note: Reviewers #1 and #2 co-reviewed your work so there are 3 reports total). You will see from their comments copied below that while they find your work of considerable potential interest, they have raised quite substantial concerns that must be addressed. In light of these comments, we cannot accept the manuscript for publication, but would be very interested in considering a revised version that addresses these serious concerns.

We found all three submitted reviews to be engaged with your study and they provide thoughtful feedback on CODEC.

Reviewers #1/#2 (co-referees) explicitly state support for publication, but have a number of high-level questions about specific aspects of CODEC.

Reviewer #3 is similarly supportive. Their comments are quite technical and make specific requests, that broadly seem doable.

Conversely, Reviewer #4 is more skeptical. While acknowledging the potential of CODEC, they identify major issues regarding the degree of overall advance; they also provide specific guidance on how to address these issues.

We note that there are multiple important points of overlap in the specific comments. For example: whether CODEC is able to detect low-frequency mutations (per Reviewers #1/#2/#4) and more technical details to show that CODEC is the advance over duplex-seq that is presented (Reviewers

#3/#4). We would like to highlight these two specific points as particularly important for a revision - they are comments that get to the heart of whether CODEC is indeed the advance presented, and we conclude that comprehensively addressing these would considerably strengthen your work.

We hope you will find the referees' comments useful as you decide how to proceed. If you wish to submit a substantially revised manuscript, please bear in mind that we will be reluctant to approach the referees again in the absence of major revisions.

To guide the scope of the revisions, the editors discuss the referee reports in detail within the team, including with the chief editor, with a view to identifying key priorities that should be addressed in revision and sometimes overruling referee requests that are deemed beyond the scope of the current study. We hope that you will find the prioritised set of referee points to be useful when revising your study. Please do not hesitate to get in touch if you would like to discuss these issues further.

If you choose to revise your manuscript taking into account all reviewer and editor comments, please highlight all changes in the manuscript text file. At this stage we will need you to upload a copy of the manuscript in MS Word .docx or similar editable format.

*2) If you have not done so already please begin to revise your manuscript so that it conforms to our Technical Report format instructions, available [here](http://www.nature.com/ng/authors/article_types/index.html). Refer also to any guidelines provided in this letter.

[redacted]

Note: This URL links to your confidential home page and associated information about manuscripts you may have submitted, or that you are reviewing for us. If you wish to forward

this email to co-authors, please delete the link to your homepage.

If you wish to submit a suitably revised manuscript we would hope to receive it within 6 months. If you cannot send it within this time, please let us know. We will be happy to consider your revision so long as nothing similar has been accepted for publication at Nature Genetics or published elsewhere. Should your manuscript be substantially delayed without notifying us in advance and your article is eventually published, the received date would be that of the revised, not the original, version.

Thank you for the opportunity to review your work.

Sincerely,

Michael Fletcher, PhD
Senior Editor, Nature Genetics

ORCID: 0000-0003-1589-7087

Referee expertise:

Referees #1/#2 (co-referees, submitted identical reports): cancer genomics, genomics methods, biomarkers.

Referee #3: cancer genomics; mutational signatures

Referee #4: genomics methods; somatic mutations

Reviewers' Comments:

Reviewers #1/#2 (co-reviewers):

Remarks to the Author:

This manuscript by Bae, Liu, et al describes a novel, clever approach to sequencing error correction (CODEC: concatenating original duplex for error correction) that requires substantially fewer reads than UMI-based duplex sequencing. CODEC uses an adapter quadruplex with strand-displacing extension, leading to the physical concatenation of both the top and bottom strands in the same

composite molecule prior to duplex dissociation. While UMI-based duplex sequencing requires high replicate rates to ensure that both strands in the duplex are adequately represented, the CODEC product ensures that both strands are contained within a single NGS molecule. If used on a routine basis, CODEC may be able to distinguish between very low VAF mutations and sequencing noise from shallow coverage NGS data.

As expected for any new methodology, there is considerable room for improvement, some aspects of which are outlined below. These limitations should be more explicitly addressed in the manuscript, with plans for how they might be improved. However, I believe the technique and accompanying pilot data are sufficiently novel and compelling to warrant publication, and I am looking forward to seeing how CODEC develops.

Major Critiques:

1. A main application of UMI-based duplex sequencing is to call low variant allele frequency (VAF) mutations within heterogeneous samples, requiring both background error correction and high-depth sampling of unique molecules. The latter is typically achieved by hybridization capture of targets of interest. Instead, the authors present many experiments with low coverage (1x) whole genome sequencing. This is effective for identifying mutation signatures but not for calling specific low-abundance mutations. (1x WGS will catch some but will miss the vast majority of mutations.) It is not clear whether CODEC is compatible with hybridization capture. This would be a major limitation.
2. Related to the above, the detection of low-abundance mutations requires high library complexity and high efficiency of adapter ligation & library preparation. From the data presented here, inefficiency of library prep appears to be an issue. Some of this is discussed in the supplement; e.g., early termination events leading to cluster phase issues, as well as various ligation isoforms leading to ~70% properly ligated molecules. There may also be ligation efficiency issues with the quadruplex adapter. Figure 2b seems to demonstrate that while duplex reads are obtained a lower read depth with CODEC, duplex discovery plateaus much earlier than duplex sequencing. Can the authors show a comparison of molecular complexity between CODEC and duplex sequencing, both with and without capture? A lower efficiency may be problematic when trying to target specific mutations from low input material, as there may be stochastic loss of the regions of interest.
3. In order to rely on single duplex molecules to call mutations, it is necessary to properly model both random and systematic contributions to the background noise profile. For instance, noise may arise from error-prone sequence context and/or particular base substitution classes. Have the authors done a detailed characterization of different sources of noise, and can that be presented here?
4. The authors claim that recovering both strands with UMI-based duplex sequencing (with dissociation) requires 100-fold excess reads. This seems like an overestimate and potentially dependent on library complexity and target size. Can the authors provide more support or context for this, as it is frequently emphasized as a selling point?

Reviewer #3:

Remarks to the Author:

In the manuscript entitled "CODEC enables 'single duplex' sequencing", Jin H Bae and colleagues

developed a new 'duplex DNA sequencing' method, CODEC. The key improvement is to concatenate 'Watson' and 'Crick' strands from a DNA fragment. This enables a tremendous reduction of sequencing throughput without substantially sacrificing the detection sensitivity. Compared to previous techniques with a similar concept (BotSeq and NanoSeq), CODEC seems to have significant advantages in the cost of sequencing. As the authors described, it has a lot of potential in cancer genomics and the field of somatic mosaicism. I believe the technique should be shared in the genomic community through a high-profile journal like Nature Genetics.

Before its publication, I have a few comments/questions/suggestions which may help to make the manuscript more meaningful.

- (1) Line 138: 72.8% of the reads showed the correct structure (presumably orientation or etc.). Please describe the pattern of discordant reads in more detail, although it is briefly introduced in SFig4. It will be essential for inspiring the genomics community to improve CODEC in the future. For example, are the discordant reads distributed randomly genome-wide or enriched in some specific regions?
- (2) In line with the first comment, I wonder how uniform the genome-wide sequencing coverage of CODEC is, compared to ordinary WGS and/or other Duplex Sequencing methods. GC or any other bias?
- (3) In BotSeq, the sequencing error rate is high in sequence ends as the authors describe in the Discussion. NanoSeq has a much lower error rate than BotSeq. What about CODEC? Is it able to show the distribution of mutation positions in the CODEC sequence reads?
- (4) By concatenating Watson and Crick strand, all the reads will have an internal 'inverted repeat structure', which may form a secondary structure. What is the impact of the structure in the sequencing and/or in the library prep?
- (5) I wonder about CODEC's detection sensitivity, at least for clonal mutations in cancers. In Fig 4a, the authors conducted some approaches. But I am not very sure. In general, breast cancers have $\sim 1/\text{Mb}$ somatic mutation rate (although there is a substantial interindividual variation), which is 3,000 per diploid genome (on average). However, standard methods (gray bars) detected only 200-800 somatic mutations. It is a bit strange to me. In addition, 1x CODEC could just cover the haploid genome at maximum, therefore, the sensitivity could not be higher than 50% of the deep (60x) whole-genome sequencing, which predominantly captures both alleles.
- (5-1) It would be helpful if the authors clearly show the association between the CODEC read depth and the mutation sensitivity (at least for clonal mutations in cancer).
- (6) For somatic mutation calling, germline genome sequences are usually necessary (most frequently peripheral blood). Does CODEC need germline genomes as well? If yes, what is the recommended sequencing method and its read depth?
- (7) The authors benchmark CODEC with ordinary WGS data analyzed by "Standard" and "Standard+MuTect2" variant callings (for example, Fig 4). I read that Standard is a more lenient method than 'Standard+Mutect2'. There are many more downstream filter steps to remove false positives in ordinary genome sequencing analysis. I am not clear about the Standard and

Standard+MuTect2 variant calling processes (I might miss it). Please describe more details about them as well.

(8) What is the minimum amount of gDNA (or # of cells) for conducting CODEC robustly?

Minor comment

In Fig 2b, why do individuals show different curves (i.e., cancer and healthy donors in CODEC)? Just experimental fluctuation?

In Fig 4c, do you find any MSI-related mutational signature?

Reviewer #4:

Remarks to the Author:

This article presents a novel duplex sequencing method called CODEC. CODEC is based on an innovative adapter structure that enables to concatenate the two strands of a DNA molecule while including all the sequencing read components and double-strand molecular tags for duplex sequencing. This produces "single duplex" reads that can be subjected to error-correction following the same principles of standard duplex sequencing. The main advantage over duplex sequencing is that it reduces read usage because the complementary strands stay together and therefore reads are not wasted to single strands. In addition, it appears to be accurate even at low family sizes, reducing sequencing needs even further. The authors present several relevant applications in which this method could be very useful by reducing sequencing costs and increasing the accuracy of current NGS approaches.

In principle, the method looks very promising. Unfortunately, the experiments designed to prove it are not clearly explained, key methodological details are missing, and the concepts of error rate and true low frequency mutations are not properly addressed. Specific critiques are listed below.

1. "Only 72.8% of the reads showed the correct structure, and 95.3% of reads retained information on at least one strand of a duplex just like standard NGS". If the % of reads showing the correct structure is so low, is there really a 'savings' in total reads compared to duplex sequencing? It seems that many reads are lost to incomplete molecules. Most importantly, having 95.3% reads with information in one strand misses the point, as that is what duplex sequencing already does.

2. What is the % of reads with correct structure in standard sonicated gDNA (72.8% is in cfDNA)? In addition, if I understand Fig. 1b correctly, when fragments are too large, read 1 and read 2 might sequence large portions of the fragment without overlapping. If so, the fragments might still have the right structure, but they are not read as "duplexes" because the sequence does not overlap. In what % of reads this occurs typically in gDNA? Please confirm and discuss in the paper whether the read must be fully traversed to have a duplex and provide information about the insert size and % of reads with correct structure. Details about insert size are not provided but are critical to evaluate this method.

3. In Fig. S1 the authors indicate that duplexes with up to 174bp can be accommodated without bending at all. How is this size calculated?

4. The authors define error rate as substitution error rate at the base level after mapping to the reference genome. While this definition might be appropriate for a sample with extremely low

mutation rate (for example blood cord or sperm, as done in the paper that the authors cite), it is not appropriate for cfDNA from a cancer patient, where multiple low frequency real mutations are expected to occur. Indeed, the "error rate" of the cancer patient is higher than the normal control. Depending on the age of the individual, the normal control is also expected to carry some true low frequency mutations. I think this is a very important problem of concept that needs to be corrected. One of the main applications of ultra-accurate sequencing is the detection of low frequency mutations. It is critical not to confuse low frequency mutations with error rate.

5. Interestingly, several paragraphs later the authors acknowledge that a subset of these mutations are not errors but biologically relevant and mention how CODEC challenges how we approach sequencing error rate calculations due to this finding. This challenge is not new and was already revealed by prior Duplex Sequencing studies and other studies reporting the presence of multiple low frequency somatic mutations in normal tissue. The authors appear to ignore this extensive body of literature in their approach to quantify error rate and the interpretation and reporting of results. The plots in Fig.2 should not be labeled "error rate" but "mutation frequency".

6. In methods, the authors mention that CODEC has 40% on-target vs 89% for Duplex Sequencing. Why is the percentage so low for CODEC? Is this average for all samples? If so, how is CODEC more efficient and cost-effective than Duplex Sequencing if more than half of the reads in a sequencing run are in off-target regions?

7. Related to the prior point, this sentence in the abstract is misleading: "CODEC affords 1000-fold higher accuracy, using 100-fold fewer reads than Duplex Sequencing." It should say: "CODEC affords 1000-fold higher accuracy than NGS, using 100-fold fewer reads than Duplex Sequencing." In addition, it is unclear from the data presented that CODEC uses 100-fold fewer reads than Duplex Sequencing. Does this estimation take into consideration the large number of reads that do not have the correct structure and the high % of reads-off target with CODEC? Also, what is the family size of the duplex sequencing data presented in Fig. 2b?

8. If I understand correctly, the authors sequenced 4 ctDNA from patients with breast cancer and 4 from normal controls but Fig. 2 shows only the data of one pair. Were the other pairs comparable? Could the authors include the data as Supplementary? Also, some key analytical details are missing. What was the pan-cancer panel used? How large is it and how many total nt were sequenced in each sample? What was the average depth for CODEC and Duplex Sequencing?

9. The cost-benefit analysis also lacks critical details about family sizes and depth for CODEC and Duplex Seq.

10. The false negative rate seems substantial (0.2) even at 5x and it is an important problem for many applications. Can it be reduced further by increasing to higher depths?

11. While it is encouraging that an average of 8.2% of mutations identified by WGS with CODEC were replicated by duplex sequencing, that leaves 91.8% of mutations not validated and suspicious of potential errors. Could the authors elaborate on that point?

12. Overall, the experiments are hard to follow and there is minimal analytical data provided to evaluate them. I'd suggest including supplementary tables in which the authors clearly indicate, for each experiment, type of sample used, input amount of DNA, total sequencing reads allocated, reads with proper CODEC structure, mean insert size, reads on target, mean family size, average depth, and number of mutations obtained.

13. The paper does not present any application of CODEC for the detection of specific low frequency mutations. CODEC appears to be very helpful for the detection of broad mutational signatures across the genome, including microsatellite instability. But it is not clear (and it is not demonstrated) whether this method has sensitivity for applications in which certain mutations must be detected as it is required for cancer diagnosis, screening, and MRD. The authors should tone down their discussion on this topic.

Author Rebuttal to Initial comments

Reviewers #1/#2 (co-reviewers):

Remarks to the Author:

This manuscript by Bae, Liu, et al describes a novel, clever approach to sequencing error correction (CODEC: concatenating original duplex for error correction) that requires substantially fewer reads than UMI-based duplex sequencing. CODEC uses an adapter quadruplex with strand-displacing extension, leading to the physical concatenation of both the top and bottom strands in the same composite molecule prior to duplex dissociation. While UMI-based duplex sequencing requires high replicate rates to ensure that both strands in the duplex are adequately represented, the CODEC product ensures that both strands are contained within a single NGS molecule. If used on a routine basis, CODEC may be able to distinguish between very low VAF mutations and sequencing noise from shallow coverage NGS data.

As expected for any new methodology, there is considerable room for improvement, some aspects of which are outlined below. These limitations should be more explicitly addressed in the manuscript, with plans for how they might be improved. However, I believe the technique and accompanying pilot data are sufficiently novel and compelling to warrant publication, and I am looking forward to seeing how CODEC develops.

Response: Thank you for the positive remarks about our study and for the helpful suggestions of ways to improve it!

Major Critiques:

1. A main application of UMI-based duplex sequencing is to call low variant allele frequency (VAF) mutations within in heterogeneous samples, requiring both background error correction and high-depth sampling of unique molecules. The latter is typically achieved by hybridization capture of targets of interest. Instead, the the authors present many experiments with low coverage (1x) whole genome sequencing. This is effective for identifying mutation signatures but not for calling specific low-abundance mutations. (1x WGS will catch some but will mist the vast majority of mutations.) It is not clear whether CODEC is compatible with hybridization capture. This would be a major limitation.

Response: We apologize for the confusion. CODEC is compatible with hybridization capture as shown in Figs. 2a-c (pan-cancer panel) and Supplementary Fig. 10 (whole-exome sequencing). This information was buried in figure legends, and we have revised the text to make it clear in the Results. Additionally, we have added a new Fig. 4h to explore the detection of specific low frequency mutations using hybridization capture on CODEC and Duplex Sequencing cell-free DNA (cfDNA) libraries. This new figure shows that CODEC uncovers tumor mutations with fewer total read pairs. Although Duplex Sequencing ultimately ‘overtakes’ CODEC at very high read counts due to its higher conversion efficiency at present, CODEC required up to 100-fold fewer read pairs to detect the tracked tumor mutations in cfDNA.

Updated text: (page 3) “..., we ran target enrichment with the IDT xGen pan-cancer hybridization capture panel (800 kb) on NGS libraries...”

(page 9) “Another potential application of CODEC is tracking mutations of interest with fewer reads. To test how CODEC improves tumor mutation detection from liquid biopsy samples, we performed hybridization capture with personalized, tumor-informed probe panels [23] on both CODEC and Duplex Sequencing libraries from four breast cancer patients’ cfDNA. Four patients with different tumor fractions showed up to 100-fold reduction on read pairs needed to detect the tracked mutations (**Fig. 4h**).”

2. Related to the above, the detection of low-abundance mutations requires high library complexity and high efficiency of adapter ligation & library preparation. From the data presented here, inefficiency of library prep appears to be an issue. Some of this is discussed in the supplement; e.g., early termination events leading to cluster phase issues, as well as various ligation isoforms leading to ~70% properly ligated molecules. There may also be ligation efficiency issues with the quadruplex adapter. Fig. 2b seems to demonstrate that while duplex reads are obtained a lower read depth with CODEC, duplex discovery plateaus much earlier than duplex sequencing. Can the authors show a comparison of molecular complexity between CODEC and duplex sequencing, both with and without capture? A lower efficiency may be problematic when trying to target specific mutations from low input material, as there may be stochastic loss of the regions of interest.

Response: We thank the reviewer for the insightful comment about molecular complexity after enrichment. To investigate your question about lower plateaus in Fig. 2b, we performed hybrid capture sequencing to saturate the duplex recovery (new Supplementary Fig. 7b). Mean molecular complexities of CODEC and Duplex Sequencing from 20 ng of four healthy donor cfDNA were found to be 3.4% and 11.6% of the original duplexes, respectively (new Supplementary Fig. 7c). We believe that with further optimization, we

may be able to bridge this 3.4x (11.6%/3.4%) gap. Unfortunately, we were not able to compare the molecular complexity without capture as the unique duplex depth of Duplex Sequencing WGS was >100-fold lower than that of CODEC WGS.

Updated text: (page 4) “Although each Duplex Sequencing library needs different amounts of sequencing depending on its molecular complexity [21], it eventually recovered more unique duplexes in an experiment where it obtained enough read pairs (Supplementary Fig. 7).”

3. In order to rely on single duplex molecules to call mutations, it is necessary to properly model both random and systematic contributions to the background noise profile. For instance, noise may arise from error-prone sequence context and/or particular base substitution classes. Have the authors done a detailed characterization of different sources of noise, and can that be presented here?

Response: Thank you for this suggestion. To investigate whether CODEC contributed to any new errors, we analyzed the trinucleotide mutation contexts of CODEC and Duplex Sequencing libraries from the same individuals in Fig. 3d. We also added a new Supplementary Fig. 15 which shows that cosine similarity between error (after excluding as many true mutations as possible) contexts of CODEC WGS and target enriched Duplex Sequencing was 0.959, suggesting that CODEC did not contribute to any new errors. The four highest peaks in each figure were cytosines in CpG contexts, and the proportions of SBS1, which reflects spontaneous deamination of cytosines, were 24.2% and 19.7% for CODEC and Duplex Sequencing, respectively.

Updated text: (page 7) “Trinucleotide context of mutations was different for each individual except high peaks at C>T mutations at CpG as expected [32] with no error specific to CODEC (Supplementary Fig. 15).”

4. The authors claim that recovering both strands with UMI-based duplex sequencing (with dissociation) requires 100-fold excess reads. This seems like an overestimate and potentially dependent on library complexity and target size. Can the authors provide more support or context for this, as it is frequently emphasized as a selling point?

Response: This is an excellent point. The exact reduction in reads required indeed depends upon the library complexity (which is related to the DNA input), the target size, and the intended depth of coverage. We have added a new Supplementary Fig. 7b, where eight new curves of unique duplex depth vs. read pairs per target were grouped by their input masses to explore this relationship. Within the same group, Duplex Sequencing starts recovering duplexes with 100-fold more reads than CODEC, but the gap closes with deeper sequencing; we have thus softened our claim as CODEC requires “up to” 100-fold fewer reads.

Updated text: (page 4) “Although each Duplex Sequencing library needs different amounts of sequencing depending on its molecular complexity [21], it eventually recovered more unique duplexes in an experiment where it obtained enough read pairs (Supplementary Fig. 7).”

Reviewer #3:

Remarks to the Author:

In the manuscript entitled "CODED enables 'single duplex' sequencing", Jin H Bae and colleagues developed a new 'duplex DNA sequencing' method, CODEC. The key improvement is to concatenate 'Watson' and 'Crick' strands from a DNA fragment. This enables a tremendous reduction of sequencing throughput without substantially sacrificing the detection sensitivity. Compared to previous techniques with a similar concept (BotSeq and NanoSeq), CODEC seems to have significant advantages in the cost of sequencing. As the authors described, it has a lot of potential in cancer genomics and the field of somatic mosaicism. I believe the technique should be shared in the genomic community through a high-profile journal like Nature Genetics.

Before its publication, I have a few comments/questions/suggestions which may help to make the manuscript more meaningful.

Response: Thank you for your kind words and helpful suggestions!

(1) Line 138: 72.8% of the reads showed the correct structure (presumably orientation or etc.). Please describe the pattern of discordant reads in more detail, although it is briefly introduced in SFig4. It will be essential for inspiring the genomics community to improve CODEC in the future. For example, are the discordant reads distributed randomly genome-wide or enriched in some specific regions?

Response: Thank you for this suggestion. We have added new Supplementary Figs. 4c,d, showing the pattern of discordant reads. We find that the discordant (i.e. “byproduct”) reads are distributed randomly throughout the genomes, suggested by the similarity between the coverage plots of correct product and byproducts (new Supplementary Fig. 4c). Interestingly, byproduct reads showed less GC bias than correct product reads. When byproduct and correct product reads were allocated to 300 bp windows with different GC contents, medians of byproduct reads were constant (new Supplementary Fig. 4d), possibly because their shorter lengths were affected less by polymerase’s GC bias. We have also updated Supplementary Figs. 4a,b to clarify the structures of CODEC’s byproducts.

(2) In line with the first comment, I wonder how uniform the genome-wide sequencing coverage of CODEC is, compared to ordinary WGS and/or other Duplex Sequencing methods. GC or any other bias?

Response: We find the genome-wide sequencing coverage of CODEC to be similar to that of standard WGS when median coverages were matched although the uniformity was slightly less (new Supplementary Fig. 9a). This is because CODEC showed more bias against fragments with higher GC content than Duplex Sequencing (new Supplementary Fig. 9b). CODEC may have been affected more by polymerase’s GC bias due to its longer fragment length.

Updated text: (page 5) “...although CODEC WGS was not as uniform as that of standard WGS at high GC content (Supplementary Fig. 9).”

(3) In BotSeq, the sequencing error rate is high in sequence ends as the authors describe in the Discussion. NanoSeq has a much lower error rate than BotSeq. What about CODEC? Is it able to show the distribution of mutation positions in the CODEC sequence reads?

Response: Yes, CODEC is also able to show the distribution of mutation positions and similarly shows higher error rates towards the sequence ends. We have added a new Supplementary Fig. 11 to show distance from fragment end and how we trim the last 12 bp (Parsons et al., *Clinical Cancer Research* 2020, Xiong et al., *Nucleic Acids Research* 2021). As our priority was to establish CODEC as a more efficient method of Duplex Sequencing, we had initially used a commercial end repair/dA-tailing (ER/AT); but we have since

added a new Fig. 2e to investigate the effect of three different ER/AT methods: commercial ER/AT, Duplex-Repair (Xiong et al., *Nucleic Acids Research* 2021), and ddBTP-blocked ER/AT (inspired by Abascal et al., *Nature* 2021). We found that up to 18.6-fold lower residual SNV frequencies were obtained when ER/AT errors were suppressed.

Updated text: (page 5) “We reasoned that some errors in CODEC sequencing result from end-repair/dA-tailing (ER/AT) as shown for Duplex Sequencing [22]. Indeed, residual mutation frequencies in CODEC were generally higher toward the ends of DNA fragments (Supplementary Fig. 11), consistent with ER/AT errors [22, 31]. To address these issues, we paired CODEC with varied ER/AT methods and applied them to sperm DNA with an expectedly low biological mutation rate. We compared a commercial ER/AT method with Duplex-Repair [31] and a custom ddBTP-blocked ER/AT inspired by Abascal et al. [22] which fully blocks ER/AT errors in theory (Supplementary Fig. 12). When we applied these to a sperm DNA (39 y/o donor) prior to CODEC, Duplex-Repair and ddBTP-blocked ER/AT showed 5.1-fold and 18.6-fold lower residual SNV frequencies (1.00×10^{-7} and 2.72×10^{-8}) than a commercial ER/AT kit (5.07×10^{-7}), respectively. (**Fig. 2e**, Supplementary Fig. 13). With ddBTP-blocked ER/AT, the result (2.72×10^{-8}) was comparable to that of the recent report (1.48×10^{-8} for 21 y/o and 4.38×10^{-8} for 73 y/o) [22].”

(4) By concatenating Watson and Crick strand, all the reads will have an internal 'inverted repeat structure', which may form a secondary structure. What is the impact of the structure in the sequencing and/or in the library prep?

Response: CODEC actually links the Watson strand with the reverse complement of the Crick strand in order to avoid an inverted repeat structure. We have clarified this point in the text and changed the shades in Fig. 1a.

Updated text: (page 3) “This strategy allowed CODEC to physically concatenate the ~~sequence information~~ of Watson strand with the reverse complement of the Crick strand into a single strand without forming a prohibitive hairpin or inverted repeat structure from two complementary sequences.”

(5) I wonder about CODEC's detection sensitivity, at least for clonal mutations in cancers. In Fig 4a, the authors conducted some approaches. But I am not very sure. In general, breast cancers have ~1/Mb somatic mutation rate (although there is a substantial interindividual variation), which is 3,000 per diploid genome

(on average). However, standard methods (gray bars) detected only 200-800 somatic mutations. It is a bit strange to me. In addition, 1x CODEC could just cover the haploid genome at maximum, therefore, the sensitivity could not be higher than 50% of the deep (60x) whole-genome sequencing, which predominantly captures both alleles.

Response: We apologize for the confusion. Fig. 4a compares 1x CODEC WGS to 1x standard WGS requiring just a single read to detect the same mutations that had been found using 60x standard WGS paired with an established mutation caller, Mutect2. We have updated the text and figure legend to clarify this. We have also added a new Supplementary Fig. 16a to show that with just 1x standard WGS and the varied tumor purities of the biopsies (median 0.39, range 0.19 - 0.69), we indeed expect to find only 200-800 mutations.

Updated text: (Fig. 4a caption) “Mutations detected by CODEC and standard WGS at 1x on eight breast tumor samples were validated by 60x standard WGS + Mutect2. Mutations on a single read were accepted.”

(page 4) “Throughout this work, we used the same variant calling pipeline for CODEC and standard NGS unless otherwise noted (See “CODEC single-fragment mutation caller” in the Methods).”

(5-1) It would be helpful if the authors clearly show the association between the CODEC read depth and the mutation sensitivity (at least for clonal mutations in cancer).

Response: We have added a new Supplementary Fig. 16b which shows sensitivity for mutations found with 60x WGS + Mutect2 at different read coverages. There was a strong linear relationship between sensitivity and coverage as expected although the theoretical sensitivity was higher assuming perfect uniformity and sequencing efficiency.

Updated text: (page 8) “With higher precision, 1x CODEC removed 99% of unvalidated mutations compared to 1x standard WGS with relatively similar validated mutation counts (Fig. 4a) while showing a linear relationship between sensitivity and coverage (Supplementary Fig. 16).”

(6) For somatic mutation calling, germline genome sequences are usually necessary (most frequently peripheral blood). Does CODEC need germline genomes as well? If yes, what is the recommended sequencing method and its read depth?

Response: Yes, removing germline mutations is necessary for somatic mutation calling. The requirements for this are no different than for other somatic analyses which may, for instance, involve sequencing a peripheral blood sample to >15x depth. We have clarified this in the text.

Updated text: (page 13) “For all patients, we sequenced matched peripheral blood cells to 15x or greater depth to omit germline polymorphisms.”

(7) The authors benchmark CODEC with ordinary WGS data analyzed by "Standard" and "Standard+MuTect2" variant callings (for example, Fig 4). I read that Standard is a more lenient method than 'Standard+Mutect2'. There are many more downstream filter steps to remove false positives in ordinary genome sequencing analysis. I am not clear about the Standard and Standard+MuTect2 variant calling processes (I might miss it). Please describe more details about them as well.

Response: Standard+Mutect2 is a “gold standard” approach for analyzing somatic mutations from tumor genomes. This entails deep (>30-60x) sequencing of the tumor biopsy and observing mutations in **multiple reads** (along with other filters) to resolve true mutations from the high number of errors in Standard NGS data. Whereas, “Standard” as presented only requires mutations to be observed in **single reads** (and is thus, more lenient), the same analysis pipeline as for CODEC mutation detection is used. We have clarified this in the text and Fig. 4 caption. To avoid confusion about sequencing depth, we changed “Standard+Mutect2” to “Mutect2”.

Updated text: (page 4) “Throughout this work, we used the same variant calling pipeline for CODEC and standard NGS unless otherwise noted (See “CODEC single-fragment mutation caller” in the Methods).”

(page 8) “CODEC and standard WGS at 1x coverage were compared by testing how well they detect mutations established by deep standard WGS (60x coverage) paired with a variant caller, Mutect2 [33], on eight breast tumor samples ~~paired with a variant caller, Mutect2 [33].~~”

(8) What is the minimum amount of gDNA (or # of cells) for conducting CODEC robustly?

Response: The experiments in the manuscript used as little as 2.5 ng of gDNA. To further explore these limits, we have performed CODEC library construction with even lower inputs of DNA to quantify the correct ratio, using Bioanalyzer High Sensitivity DNA Chip. We have included these data as a new Supplementary Fig. 19 which suggests the potential to use CODEC with as little as 10-100 pg of DNA.

Updated text: (page 11) “Input mass for CODEC ranged from 2.5 to 20 ng although we observed the correct product with as little as 0.01 ng mass (Supplementary Fig. 19).”

Minor comment

In Fig 2b, why do individuals show different curves (i.e., cancer and healthy donors in CODEC)? Just experimental fluctuation?

Response: The shape of the curve is dictated by the molecular complexity and uniformity of the sequencing library, and it is known to vary (Parsons et al., Clinical Cancer Research 2020, Cohen et al., Nature Biotechnology 2021). To further explore this, we have added a new Supplementary Fig. 7b to show how these curves vary between additional individuals.

In Fig 4c, do you find any MSI-related mutational signature?

Response: Yes, COSMIC MSI signatures are indicated in green in Fig. 4e.

Reviewer #4:

Remarks to the Author:

This article presents a novel duplex sequencing method called CODEC. CODEC is based on an innovative adapter structure that enables to concatenate the two strands of a DNA molecule while including all the sequencing read components and double-strand molecular tags for duplex sequencing. This produces “single duplex” reads that can be subjected to error-correction following the same principles of standard duplex sequencing. The main advantage over duplex sequencing is that it reduces read usage because the complementary strands stay together and therefore reads are not wasted to single strands. In addition, it appears to be accurate even at low family sizes, reducing sequencing needs even further. The authors present several relevant applications in which this method could be very useful by reducing sequencing costs and increasing the accuracy of current NGS approaches.

In principle, the method looks very promising. Unfortunately, the experiments designed to prove it are not clearly explained, key methodological details are missing, and the concepts of error rate and true low frequency mutations are not properly addressed. Specific critiques are listed below.

Response: Thank you for your kind words and helpful critiques!

1. “Only 72.8% of the reads showed the correct structure, and 95.3% of reads retained information on at least one strand of a duplex just like standard NGS”. If the % of reads showing the correct structure is so low, is there really a ‘savings’ in total reads compared to duplex sequencing? It seems that many reads are lost to incomplete molecules. Most importantly, having 95.3% reads with information in one strand misses the point, as that is what duplex sequencing already does.

Response: We apologize for the confusion. We meant that 72.8% of the reads retained information of both strands, 22.5% of the reads retained information of only one strand, and their sum (which is 95.3% of the reads) retained information of either one or both strands. Yet, as you have indicated, reads with information of only one strand aren’t relevant to the point. We have thus removed this point from the paper. In addition, Supplementary Figs. 4a,b now show clearer detail on the structures of byproducts to clarify what the “correct structure” means.

Updated text: (page 3) “We found that 72.8% of the reads showed the correct structure and 95.3% of reads retained information from both strands ~~on at least one of a duplex just like standard NGS~~ (Supplementary Fig. 4).”

2. What is the % of reads with correct structure in standard sonicated gDNA (72.8% is in cfDNA)?

Response: Among 16 sonicated gDNA samples, 61.7% (range 55.7 – 73.3%) of the CODEC WGS reads showed the correct structure. We have also recomputed the percentage for cfDNA including the new samples added in our revised paper (now, eight in total) and found it to be 68.4% (range 59.7 – 75.4%). This information is now included in a new Supplementary Table 1.

In addition, if I understand Fig. 1b correctly, when fragments are too large, read 1 and read 2 might sequence large portions of the fragment without overlapping. If so, the fragments might still have the right structure, but they are not read as “duplexes” because the sequence does not overlap. In what % of reads this occurs typically in gDNA? Please confirm and discuss in the paper whether the read must be fully traversed to have a duplex and provide information about the insert size and % of reads with correct structure. Details about insert size are not provided but are critical to evaluate this method.

Response: We apologize for the missing detail. Each full duplex does not need to be fully traversed to determine the structure because Read 1 and 2 are individually aligned to the reference first. We updated the manuscript to make it clear and have added a new Supplementary Fig. 5 to explore how (a) correct product ratio and (b) the percentage of base pairs sequenced from both strands, varies with the mean insert size of the library. Among the same sample type, longer insert size did not negatively affect correct reads or bases as expected. However, we agree that details about insert size are critical and have disclosed them in new Supplementary Table 1.

Updated text: (page 3) “Because each insert does not need to be fully sequenced to determine the structure (Supplementary Notes), the ratio of correct reads was not negatively affected by longer insert size (Supplementary Fig. 5). More details on each NGS dataset can be found in Supplementary Table 1.”

3. In Fig. S1 the authors indicate that duplexes with up to 174bp can be accommodated without bending at all. How is this size calculated?

Response: This size was calculated using the lengths of DNA in B-DNA helix and single-stranded structure. Approximately, it is 0.33 nm per base pair along the helical axis of B-DNA and 0.64 nm per nucleotide for single-stranded DNA. We excluded 3 nucleotides from each single-stranded region (green

and yellow) because we conservatively assumed the minimum length to reverse the direction of a single-stranded DNA strand is the same as the minimum length of a hairpin loop. For instance, if the length of an insert is N base pairs,

$$(2 \times 19 \text{ bp} + N \text{ bp}) \times 0.33 \text{ nm/bp} = 2 \times (50-3) \text{ nt} \times 0.64 \text{ nm/nt} + 30 \text{ bp} \times 0.33 \text{ nm/bp}$$

Thus, $N = 174.3$

Update text: (Supplementary Fig. 1b caption) “Duplexes with up to 174 bp can be accommodated without bending at all, which was calculated using the lengths of DNA in B-DNA helix and single-stranded structure. Approximately, it is 0.33 nm per base pair along the helical axis of B-DNA and 0.64 nm per nucleotide for single-stranded DNA. We excluded 3 nucleotides from each single-stranded region (green and yellow) because we conservatively assumed the minimum length to reverse the direction of a single-stranded DNA strand is the same as the minimum length of a hairpin loop. For instance, if the length of an insert is N base pairs, $(2 \times 19 \text{ bp} + N \text{ bp}) \times 0.33 \text{ nm/bp} = 2 \times (50-3) \text{ nt} \times 0.64 \text{ nm/nt} + 30 \text{ bp} \times 0.33 \text{ nm/bp}$. Thus, $N = 174.3$.”

4. The authors define error rate as substitution error rate at the base level after mapping to the reference genome. While this definition might be appropriate for a sample with extremely low mutation rate (for example blood cord or sperm, as done in the paper that the authors cite), it is not appropriate for cfDNA from a cancer patient, where multiple low frequency real mutations are expected to occur. Indeed, the “error rate” of the cancer patient is higher than the normal control. Depending on the age of the individual, the normal control is also expected to carry some true low frequency mutations. I think this is a very important problem of concept that needs to be corrected. One of the main applications of ultra-accurate sequencing is the detection of low frequency mutations. It is critical not to confuse low frequency mutations with error rate.

Response: We appreciate this concern. Based upon this comment and the reviewer’s subsequent suggestion, we have opted to use “residual SNV frequency” and “residual indel frequency” instead of “error rate.” The rationale is that we are computing the number of base changes left after (1) mapping to the reference genome, and (2) excluding high-abundance germline and somatic SNVs and indels detected with methods such as Haplotype Caller and Mutect2. We have clarified this definition and updated all figures and captions accordingly.

Updated text: (page 3) “Of note, we use “residual SNV frequency” or “residual indel frequency” instead of “error rate” to reflect the mutations left after excluding the germline and high-abundance somatic mutations [20, 22, 29] (Methods).”

5. Interestingly, several paragraphs later the authors acknowledge that a subset of these mutations are not errors but biologically relevant and mention how CODEC challenges how we approach sequencing error rate calculations due to this finding. This challenge is not new and was already revealed by prior Duplex Sequencing studies and other studies reporting the presence of multiple low frequency somatic mutations in normal tissue. The authors appear to ignore this extensive body of literature in their approach to quantify error rate and the interpretation and reporting of results. The plots in Fig.2 should not be labeled “error rate” but “mutation frequency”.

Response: We did not mean to suggest that finding rare somatic mutations was a new discovery. We have removed the sentence “It also challenges how we approach sequencing error rate calculations...” and have changed all instances of “error rate” to “residual SNV/indel frequency” as described above, including the plots in Fig. 2. Additionally, we have clarified how prior studies (e.g. Schmitt et al., PNAS 2012, Abascal et al., Nature 2021, Moore et al., Nature 2021) have revealed the presence of true, low frequency somatic mutations.

Updated text: (page 3) “Of note, we use “residual SNV frequency” or “residual indel frequency” instead of “error rate” to reflect the mutations left after excluding the germline and high-abundance somatic mutations [20, 22, 29] (Methods).”

~~(page 8) “It also challenges how we approach sequencing error rate calculations (e.g., Figs. 2acd, 3bc, 4a) having found that a non-trivial subset of apparent ‘errors’ in CODEC sequencing could be validated as true, low abundance somatic mutations.”~~

6. In methods, the authors mention that CODEC has 40% on-target vs 89% for Duplex Sequencing. Why is the percentage so low for CODEC?

Response: To our understanding, the lower enrichment factor is in part because the current hybridization capture protocol is designed for standard NGS libraries, whereas we have not yet optimized blockers for

CODEC including its unique linker sequence. Indeed, the on-target rates that we observed for CODEC are consistent with those reported for standard NGS libraries when adapter blockers are not used (see following figure from Twist Bioscience, where x-axis shows targeted region length).

(<https://www.twistbioscience.com/products/ngs/reagent-kits/universal-blockers?tab=data>). Of note, 40% is the on-target percentage for correct CODEC reads after a single capture, but any byproducts with the on-target sequence will further reduce this percentage. We have recomputed the on-target percentage including byproduct reads (28.8%) and double capture data to disclose them in the Discussion.

Updated text: (page 10) “Applications involving hybridization capture could be further enhanced by achieving higher on-target ratio. After one or two rounds of hybridization capture [41] on cfDNA (Supplementary Table 1), mean on-target ratios of CODEC were 28.8% and 71.2%, respectively (89.0% and 99.2% for Duplex Sequencing), which we believe can be improved by using a longer hybridization blocker with LNA like commercial blockers for the CODEC linker.”

Is this average for all samples? If so, how is CODEC more efficient and cost-effective than Duplex Sequencing if more than half of the reads in a sequencing run are in off-target regions?

Response: This was the average from the two samples shown in Fig. 2b, but we have since added eight more samples with a double hybridization capture protocol (Schmitt et al., Nature Methods 2015) that improved on-target ratios (Supplementary Table 1). After including byproducts to on-target ratio calculation, means of single capture and double capture data were 28.8% and 71.2%, respectively. Even with lower on-target ratio (CODEC 71.2% vs. Duplex Seq 99.2% after double capture), we believe that up to 100-fold fewer read pairs per target required by CODEC still offsets such loss and remains as a significant improvement.

Updated text: (page 10) “Applications involving hybridization capture could be further enhanced by achieving higher on-target ratio. After one or two rounds of hybridization capture [41] on cfDNA (Supplementary Table 1), mean on-target ratios of CODEC were 28.8% and 71.2%, respectively (89.0% and 99.2% for Duplex Sequencing), which we believe can be improved by using a longer hybridization blocker with LNA like commercial blockers for the CODEC linker.”

7. Related to the prior point, this sentence in the abstract is misleading: “CODEC affords 1000-fold higher accuracy, using 100-fold fewer reads than Duplex Sequencing.” It should say: “CODEC affords 1000-fold higher accuracy than NGS, using 100-fold fewer reads than Duplex Sequencing.”

Response: Thanks for the rework of this sentence; we have made this change.

Updated text: (Abstract) “CODEC affords 1000-fold higher accuracy than NGS, using up to 100-fold fewer reads than Duplex Sequencing.”

In addition, it is unclear from the data presented that CODEC uses 100-fold fewer reads than Duplex Sequencing. Does this estimation take into consideration the large number of reads that do not have the correct structure and the high % of reads-off target with CODEC?

Response: Thank you for bringing this to our attention. Fig. 2b now shows that after target enrichment with hybridization capture, CODEC requires up to 230-fold fewer reads than Duplex Sequencing with byproducts and on-target ratios taken into account. Fig. 2d summarizing WGS shows that CODEC costs 60-fold less, also with byproducts taken into account. Yet, recognizing that the exact fold reduction in reads required may also vary depending upon library complexity, target size, and depth of sequencing, we have further added a new Supplementary Fig. 7b to explore these relationships.

Updated text: (page 3) “For a fair comparison, we used raw sequencing coverage to include CODEC byproducts throughout this work.”

(page 4) “In contrast, CODEC started to recover them with ~~350~~ 230-fold fewer read pairs despite byproducts and off-target reads.”

(page 5) “In a cost-benefit analysis, the cost of CODEC was ~~100~~ 60 times lower than that of Duplex Sequencing while maintaining higher accuracy than standard WGS (Fig. 2d)...”

Also, what is the family size of the duplex sequencing data presented in Fig. 2b?

Response: The minimum required reads of each strand was two for Duplex Sequencing, consistent with the literature (Abascal et al., Nature 2021, Parsons et al., Clinical Cancer Research 2020). This is a relaxed version of the original, which required three reads of each strand (Schmitt et al., PNAS 2012, Loeb et al., PNAS 2019). We have also added a new Supplementary Fig. 8 that shows less stringent duplex recovery with only one read per strand required. Our new Supplementary Table 1 now includes the mean family sizes observed from NGS data.

Updated text: (page 3) “The minimum required reads of each strand was two for Duplex Sequencing [22, 23].”

(Fig. 2a caption) “Duplex Sequencing required at least two reads of each strand [22].”

8. If I understand correctly, the authors sequenced 4 ctDNA from patients with breast cancer and 4 from normal controls but Fig. 2 shows only the data of one pair. Were the other pairs comparable? Could the authors include the data as Supplementary?

Response: Unlike Fig. 2 which involved target enrichment to compare with targeted Duplex Sequencing for one pair of samples, the next batch of cfDNA samples (from 4 additional patients with breast cancer and 4 healthy donors) had originally only been sequenced with shallow whole-genome sequencing. We have since performed hybrid capture sequencing on these samples too and found comparable results (new Supplementary Fig. 7a,b).

Also, some key analytical details are missing. What was the pan-cancer panel used? How large is it and how many total nt were sequenced in each sample? What was the average depth for CODEC and Duplex Sequencing?

Response: For Fig. 2, hybrid capture was performed using the IDT xGen Pan-cancer Hyb Panel (targeting 800 kb). Mean total nt sequenced for CODEC and Duplex Sequencing were 1.30×10^{10} and 7.74×10^9 , respectively. Mean unique duplex depths for CODEC and Duplex Sequencing was 44.4 and 92.6, respectively. These details are now included in a new Supplementary Table 1.

Updated Text: (page 3) "...we ran target enrichment with the IDT xGen pan-cancer hybridization capture panel (800 kb) on NGS libraries,..."

9. The cost-benefit analysis also lacks critical details about family sizes and depth for CODEC and Duplex Seq.

Response: For Fig. 2d, because whole-genome Duplex Sequencing could not recover any duplexes with the standard threshold of two or three reads of each strand (Abascal et al., Nature 2021, Parsons et al., Clinical Cancer Research 2020, Schmitt et al., PNAS 2012), we relaxed it only for this analysis to one read of each strand. We have clarified this in the text. We have also added a new Supplementary Table 1 that summarizes key NGS metrics including mean family sizes and duplex depths, which have also been added to Fig. 2d. Additionally, we have added a new Supplementary Fig. 8 exploring the impact of a less stringent family size requirement of just one read per strand for hybrid capture Duplex Sequencing.

Updated text: (Fig. 2d caption and page 4) "Because Duplex Sequencing WGS could not recover any duplex with the standard threshold, we had to relax it to one read of each strand only for this analysis."

10. The false negative rate seems substantial (0.2) even at 5x and it is an important problem for many applications. Can it be reduced further by increasing to higher depths?

Response: To address this comment, we prepared and analyzed 17x CODEC and standard WGS on NA12878. The false negative rates of both CODEC and standard WGS indeed dropped to 0.057 and 0.017,

respectively, whereas the false positives per Mb were 11.5 and 82.0, respectively (Supplementary Table 2).

Updated text: (page 5) “At 17x coverage, the FN rate of CODEC was reduced to 0.057 (Supplementary Table 2). Although there was no longer a significant advantage at the higher coverage because the FP per million of standard WGS also dropped to 82, it was still 7.1-fold higher than that of CODEC.”

11. While it is encouraging that an average of 8.2% of mutations identified by WGS with CODEC were replicated by duplex sequencing, that leaves 91.8% of mutations not validated and suspicious of potential errors. Could the authors elaborate on that point?

Response: We believe that the cross-validation ratio of 8.2% is not a surprising result under the experiment setting with the average unique depth of 2311 duplexes. A new Supplementary Fig. 14b shows the relationship between variant allele frequency (VAF) and theoretical cross-validation probability based on the binomial distribution. Considering most somatic mutations in buffy coat DNA aren't under positive selection pressure (Martincorena et al., Cell 2017) and thus have low VAF, we think that this figure explains why only a subset of mutations identified by CODEC WGS appeared again in the independent libraries.

Updated text: (page 6) “We estimate that more could be found to be true somatic mutations at lower abundance if we were to sequence even deeper (Supplementary Fig. 14b).”

12. Overall, the experiments are hard to follow and there is minimal analytical data provided to evaluate them. I'd suggest including supplementary tables in which the authors clearly indicate, for each experiment, type of sample used, input amount of DNA, total sequencing reads allocated, reads with proper CODEC structure, mean insert size, reads on target, mean family size, average depth, and number of mutations obtained.

Response: Thank you for this suggestion. We have added a new Supplementary Table 1 with all of this information for each experiment.

13. The paper does not present any application of CODEC for the detection of specific low frequency mutations. CODEC appears to be very helpful for the detection of broad mutational signatures across the genome, including microsatellite instability. But it is not clear (and it is not demonstrated) whether this method has sensitivity for applications in which certain mutations must be detected as it is required for cancer diagnosis, screening, and MRD. The authors should tone down their discussion on this topic.

Response: We have added a new Fig. 4h to explore the detection of specific low frequency mutations using hybridization capture on CODEC and Duplex Sequencing cfDNA libraries. This figure shows that CODEC uncovers tumor mutations with fewer read pairs. Although Duplex Sequencing ultimately ‘overtakes’ CODEC at very high read counts due to its higher conversion efficiency at present, CODEC required up to 100-fold fewer read pairs to discover the tracked tumor mutations.

Updated text: (page 9) “Another potential application of CODEC is tracking mutations of interest with fewer reads. To test how CODEC improves tumor mutation detection from liquid biopsy samples, we performed hybridization capture with personalized, tumor-informed probe panels [23] on both CODEC and Duplex Sequencing libraries from four breast cancer patients’ cfDNA. Four patients with different tumor fractions showed up to 100-fold reduction on read pairs needed to detect the tracked mutations (**Fig. 4h**).”

Decision Letter, first revision:

13th Sep 2022

Dear Viktor,

Your Technical Report, "CODEC enables 'single duplex' sequencing" has now been seen by 4 referees. You will see from their comments below that while they continue to find your work of interest and say it has improved in revision, some important points still remain to be addressed. We are interested in the possibility of publishing your study in Nature Genetics, but would like to consider your response to these concerns in the form of a revised manuscript before we make a final decision on publication.

In brief, the co-Referees #1 and #2 are now satisfied and have no further comments; Reviewers #3 and #4, however, still have some issues. Most of their requests appear minor and do not seem to require substantial new work, apart from Reviewer #3's comment #1 on cancer genome sensitivity; however, we leave it up to you and your co-authors to judge the most appropriate response to this request.

To guide the scope of the revisions, the editors discuss the referee reports in detail within the team, including with the chief editor, with a view to identifying key priorities that should be addressed in

revision and sometimes overruling referee requests that are deemed beyond the scope of the current study. We hope that you will find the prioritized set of referee points to be useful when revising your study. Please do not hesitate to get in touch if you would like to discuss these issues further.

We therefore invite you to revise your manuscript taking into account all reviewer and editor comments. Please highlight all changes in the manuscript text file. At this stage we will need you to upload a copy of the manuscript in MS Word .docx or similar editable format.

*2) If you have not done so already please begin to revise your manuscript so that it conforms to our Technical Report format instructions, available [here](http://www.nature.com/ng/authors/article_types/index.html). Refer also to any guidelines provided in this letter.

[redacted]

We hope to receive your revised manuscript within four to eight weeks. If you cannot send it within this time, please let us know.

Nature Genetics is committed to improving transparency in authorship. As part of our efforts in this direction, we are now requesting that all authors identified as 'corresponding author' on published

papers create and link their Open Researcher and Contributor Identifier (ORCID) with their account on the Manuscript Tracking System (MTS), prior to acceptance. ORCID helps the scientific community achieve unambiguous attribution of all scholarly contributions. You can create and link your ORCID from the home page of the MTS by clicking on 'Modify my Springer Nature account'. For more information please visit www.springernature.com/orcid.

Sincerely,

Michael Fletcher, PhD
Senior Editor, Nature Genetics

ORCID: 0000-0003-1589-7087

Referee expertise:

Referees #1/#2 (co-referees): cancer genomics, genomics methods, biomarkers.

Referee #3: cancer genomics; mutational signatures

Referee #4: genomics methods; somatic mutations

Reviewers' Comments:

Reviewer #1:

Remarks to the Author:

The authors have addressed all of my critiques and have strengthened the manuscript. While not fully mature, the method is novel and interesting and worthy of publication.

Reviewer #2:

Remarks to the Author:

No additional comments. The authors have sufficiently addressed my questions from the initial review.

Brian Loomis

Reviewer #3:

Remarks to the Author:

The revised version of the manuscript, entitled CODEC enables 'single duplex' sequencing is

substantially improved from the original one. Many of my questions are now resolved, and I appreciate the authors' efforts. Below are some additional suggestions which may help the manuscript become more clarified and meaningful.

- (1) The authors show the accuracy of CODEC even from low-pass CODEC sequencing (<5x). The precision shown by many different but complementary parameters (i.e., Residual SNV frequency in Fig 2, False Positive in Figs 3, 4, and cosine similarity of mutational signatures) looks great. However, 'detection sensitivity' for (clonal) mutations is also important, particularly in cancer genome sequencing. Although the authors suggest that 17x coverage can detect ~94% of (germline) polymorphisms (page5), I am wondering about the sequencing coverage of CODEC to detect the vast majority of somatic mutations that were identified by ordinary NGS sequencing (30-60x). Probably, the authors may apply high-depth CODEC sequencing to the breast tumor samples to show the sensitivity of CODEC over its sequencing depth.
- (2) I am wondering about the genome-wide distribution of the CODEC byproducts. Are there any particular genomic regions with higher fractions of CODEC byproducts than the background?
- (3) Fig 2a; It will be helpful for general readers if the authors graphically show what "No consensus", "R1+R2", and "SSC" mean.
- (4) Fig 3b. y-axis for the bottom panel. FP per Million. FP per Million bases? reads? Please clarify.
- (5) Fig 2c. The linear correlation with age seems great, suggesting many of the calls are real mutations. However, I am wondering about the estimated fraction of false positive calls for each dot.
- (6) Fig S5 right panel. What is the meaning of the Base Sequenced from Both Strands?

Reviewer #4:

Remarks to the Author:

The authors have made a substantial effort to address the reviewers concerns and the manuscript has greatly improved from its prior version. There are still, however, several points that remain confusing. It would be ideal if the authors could further revise the manuscript to clarify.

1. The limitation of CODEC to recover molecular complexity could be better presented and discussed. The issue is addressed in the new Suppl Fig. S7b, which should probably replace Fig. 2b, since it is more informative.
2. Throughout the paper, it is still unclear what is the mean depth of sequencing in many experiments. Could the authors please add that information in the supplementary tables?
3. The much lower sequencing requirement of CODEC vs Duplex Sequencing is unclear in view of the Supplementary Table data. For samples in figure S7 (cfDNA from BC and HD patients), the total read pair sequenced with CODEC are about half of the read pairs used with Duplex Sequencing. So, overall, it looks like CODEC, as presented, needs half the sequencing requirement of Duplex Seq, not 100-fold. This should be discussed or clarified.
4. In Fig. 4h, could the authors please indicate whether the mutations identified were all the mutations targeted for the given tumors and, if not, what proportion was missed?
5. This sentence is unclear: "Although single strand consensus (SSC) was more accurate than the no

consensus and paired-end reads consensus (R1+R2), without a consensus of Watson and Crick strands, its residual mutation frequency error rate was still 100 234-fold higher than that of CODEC.” Do the authors mean that SSC has more error than CODEC? But that is irrelevant since the comparison should be at the duplex level, right? In addition, the residual mutation frequency might not be necessarily error but low frequency biological mutations?

Author Rebuttal, first revision:

Reviewer #3:

Remarks to the Author:

The revised version of the manuscript, entitled CODEC enables 'single duplex' sequencing is substantially improved from the original one. Many of my questions are now resolved, and I appreciate the authors' efforts. Below are some additional suggestions which may help the manuscript become more clarified and meaningful.

Response: Thank you for the additional helpful suggestions of ways to clarify the manuscript!

(1) The authors show the accuracy of CODEC even from low-pass CODEC sequencing (<5x). The precision shown by many different but complementary parameters (i.e., Residual SNV frequency in Fig 2, False Positive in Figs 3, 4, and cosine similarity of mutational signatures) looks great. However, 'detection sensitivity' for (clonal) mutations is also important, particularly in cancer genome sequencing. Although the authors suggest that 17x coverage can detect ~94% of (germline) polymorphisms (page5), I am wondering about the sequencing coverage of CODEC to detect the vast majority of somatic mutations that were identified by ordinary NGS sequencing (30-60x). Probably, the authors may apply high-depth CODEC sequencing to the breast tumor samples to show the sensitivity of CODEC over its sequencing depth.

Response: We greatly appreciate this suggestion. We have sequenced the existing CODEC tumor library from Breast Cancer Patient #21 to 9-fold higher coverage (18x, duplex depth 3.7x) and found 21-fold more clonal tumor mutations (now, 71% sensitivity, as expected) from 3.7x of single duplex reads. Yet, if we apply a standard mutation caller such as Mutect2 (Benjamin *et al.*) to the full 18x sequencing of the CODEC library (including all CODEC byproducts and reads that do not pass CODEC filters), this effectively becomes standard NGS, and we uncover 94% of the clonal mutations. This underscores the versatility of CODEC: all reads can be used for standard analyses such as clonal mutation detection whereas correctly linked single duplex reads can be used to uncover the rare mutations and signatures otherwise obscured in standard NGS.

We have clarified this point in the text; revised Fig. 4a to focus on the high precision of single duplex reads; and moved our old Fig. 4a to Supplementary Fig. 17a and added a new Supplementary Fig. 17b to show clonal mutation detection using ‘single duplex’ versus all reads. We have also clarified our definitions of sequencing coverage throughout and included all detailed NGS metrics in Supplementary Table 1. For instance, for the germline polymorphism example noted above, 17x was “correct product depth” whereas the deduplicated coverage was 40x. If we analyze the full 40x with established tools, 97.3% are detected, further demonstrating CODEC’s versatility. This is all now clarified in the paper.

Updated text: (page 3) “For a fair comparison throughout this work, we matched CODEC and standard NGS on raw read pairs and deduplicated sequencing coverage for targeted and WGS, respectively, as per standard conventions [29]. Raw and deduplicated coverages include all CODEC byproduct reads and bases that do not pass all CODEC filters (Methods), representing the cost of the sequencing. Unless otherwise indicated, coverage (e.g. 30x) refers to deduplicated coverage, whereas correct product depths are calculated based on deduplicated CODEC reads with the correct structure, and duplex depths are computed after all CODEC filters are implemented (Methods). Duplex depths for CODEC will thus always be lower than the indicated coverages. However, all raw or deduplicated reads can still be utilized for standard NGS analyses such as clonal mutation detection with established tools.

(page 5) “At 40x coverage (17x correct product depth), the FN rate was further reduced to 0.057 (0.026 using full 40x coverage including byproducts, Supplementary Table 2).”

(page 7) “With higher accuracy, 2x CODEC (range 1.4-2.4x, duplex depth 0.11-0.14x) had 83 times higher fraction of SNV validated by 60x standard WGS + Mutect2 than 2x standard WGS (**Fig. 4a**, Supplementary Fig. 17a), and increasing CODEC coverage improved sensitivity as expected (Supplementary Fig. 17b).”

(page 10) “On the analysis side, a modified pipeline that salvages discarded reads and bases may prevent data loss. For example, treating byproducts as standard NGS data to supplement results from CODEC data may utilize all sequencing information.”

(2) I am wondering about the genome-wide distribution of the CODEC byproducts. Are there any particular genomic regions with higher fractions of CODEC byproducts than the background?

Response: To directly examine whether there are particular genomic regions with higher fractions of CODEC byproducts, we added a new Supplementary Fig. 4e using ichorCNA (Adalsteinsson *et al.*, *Nat*

Commun. 2017) to calculate the ratio of GC-corrected read counts per 50kb bin normalized by the LOESS-fitted (by GC) chromosome-wide mean value. When compared to the standard NGS data, CODEC byproducts showed lower normalized values, suggesting that no particular genomic region is significantly more responsible for the byproducts. Along with Supplementary Figs. 4c-d, we believe that this new analysis shows the random origin of CODEC byproducts.

(3) Fig 2a; It will be helpful for general readers if the authors graphically show what "No consensus", "R1+R2", and "SSC" mean.

Response: Thank you for suggesting a good way to clarify the manuscript. We have added a new Supplementary Fig. 7 to graphically describe what each consensus means.

(4) Fig 3b. y-axis for the bottom panel. FP per Million. FP per Million bases? reads? Please clarify.

Response: It is FP per million bases, and we edited the y-axis label of Fig. 3b accordingly.

(5) Fig 2c. The linear correlation with age seems great, suggesting many of the calls are real mutations. However, I am wondering about the estimated fraction of false positive calls for each dot.

Response: Although the fraction of false calls at low mutation level can be greatly influenced by the choice of ground truth, we compared the linear regression result of CODEC data in Fig. 3c to that of Abascal *et al.* (*Nature* 2021) (figure below). The fraction of false positives was estimated to be the distance between a data point and fitted line by Abascal *et al.* (red) divided by its y-value. Such fractions in CODEC data ranged from 17-30%, depending on ages of patients. Of note, this might not be a fair comparison because Abascal *et al.* were able to include individuals of age 0 to the linear regression model, whereas we extrapolated our data to estimate the y-intercept.

(6) Fig S5 right panel. What is the meaning of the Base Sequenced from Both Strands?

Response: We meant the final percentage of bases that passed all analytical filters and successfully generated double strand consensus. To clarify, the y-axis label and the caption of Supplementary Fig. 5b has been changed to 'post-filter bases / total bases (%)'.

Reviewer #4:

Remarks to the Author:

The authors have made a substantial effort to address the reviewers concerns and the manuscript has greatly improved from its prior version. There are still, however, several points that remain confusing. It would be ideal if the authors could further revise the manuscript to clarify.

Response: Thank you for the additional insightful comments that helped clarify the manuscript!

1. The limitation of CODEC to recover molecular complexity could be better presented and discussed. The issue is addressed in the new Suppl Fig. S7b, which should probably replace Fig. 2b, since it is more informative.

Response: We followed your advice and merged Fig. 2b and old Supplementary Fig. 7b (which is now Supplementary Fig. 8b). For a fair comparison, data from input mass over 20 ng or below 8 ng were excluded because they were sequenced only with either CODEC or Duplex Sequencing, but not both. We also added a sentence to directly acknowledge the fact that CODEC has lower library conversion efficiency.

Updated text: (page 4) “When we sequenced more cfDNA samples of healthy donors and breast cancer patients, we found that Duplex Sequencing could not start recovering original duplexes until receiving 600 read pairs per target on average (CODEC median 25M raw read pairs, range 16M-54M; Duplex Sequencing median 31M, range 19M-72M) (**Fig. 2c**). In contrast, CODEC started to recover them with 220-fold fewer read pairs despite byproducts and off-target reads.”

(page 4) “On the other hand, its lower plateau was probably due to lower library conversion efficiency. Although each Duplex Sequencing library needs different amounts of sequencing depending on its molecular complexity [21], Duplex Sequencing eventually recovered more unique duplexes when it obtained enough read pairs (See Supplementary Fig. 8 for full result).”

2. Throughout the paper, it is still unclear what is the mean depth of sequencing in many experiments. Could the authors please add that information in the supplementary tables?

Response: Thank you for this comment. We have extensively revised the paper to clarify our definitions of sequencing coverages and have added the sequencing coverages inline to each section of the Results and Supplementary Table 1. For each sample, we report raw NGS read pairs, deduplicated NGS coverage including byproducts, correct product depth, and duplex depth reflecting correct product CODEC reads passing all analysis filters. For all analyses throughout the paper, CODEC is matched to standard NGS data using raw or deduplicated reads for targeted or WGS, respectively, both of which include all CODEC byproducts. This allows for a fair and conservative evaluation of CODEC.

Further, we have carefully checked every single analysis result in the paper for consistency in read matching and have made a series of minor corrections with negligible impact on results. This has incurred some late-stage changes to the manuscript, and we apologize for the inconvenience.

- Fig. 2d: cost of CODEC was reduced from \$0.139/Mbp to \$0.097/Mbp when using the cost of raw read pairs like the other cost analyses

- Fig. 3b: False positive per million base sequenced and false negative rate of CODEC on NA12878 germline mutation detection were decreased by 23% and increased by 51% on average, respectively, when we matched its deduplicated coverage to that of standard WGS (previously, byproducts had been excluded, but this is now fixed)
- Fig. 3c: CODEC R^2 was decreased from 0.81 to 0.80 when we downsampled coverages of two samples that previously had higher coverages (from 18x to 6x)
- Fig. 4a: now presents fractions of mutations validated to put more focus on precision (old Fig. 4a was moved to Fig. S17a)
- Fig. 4f: cosine similarities of CODEC were decreased from 0.89 to 0.87 on average when we matched its deduplicated coverage to that of standard WGS (previously, byproducts had been excluded, but this is now fixed)
- Fig. 4g: Pearson's r of CODEC was increased from 0.775 to 0.910, whereas that of standard WGS were decreased from -0.107 to -0.12, when we matched CODEC's deduplicated coverage to that of standard WGS (previously, byproducts had been excluded, but this is now fixed)
- Fig. S17a: CODEC mutations validated and unvalidated by 60x standard + Mutect2 were decreased by 38% and 54% on average, respectively, when we matched its deduplicated coverage to that of standard WGS (previously, byproducts had been excluded, but this is now fixed)
- Fig. S17b: replaced multiple low-coverage data with a single high-coverage data as per Reviewer 3's request

We now assure that all data is precisely matched and that all prior conclusions hold.

Updated text: (page 3) “For a fair comparison throughout this work, we matched CODEC and standard NGS on raw read pairs and deduplicated sequencing coverage for targeted and WGS, respectively, as per standard conventions [29]. Raw and deduplicated coverages include all CODEC byproduct reads and bases that do not pass all CODEC filters (Methods), representing the cost of the sequencing. Unless otherwise indicated, coverage (e.g. 30x) refers to deduplicated coverage, whereas correct product depths are calculated based on deduplicated CODEC reads with the correct structure, and duplex depths are computed after all CODEC filters are implemented (Methods). Duplex depths for CODEC will thus always be lower than the indicated coverages. However, all raw or deduplicated reads can still be utilized for standard NGS analyses such as clonal mutation detection with established tools.”

(page 4) “When we sequenced more cfDNA samples of healthy donors and breast cancer patients, we found that Duplex Sequencing could not start recovering original duplexes until receiving 600 read pairs per target on average (CODEC median 25M raw read pairs, range 16M-54M; Duplex Sequencing median 31M, range 19M-72M) (Fig. 2c).”

(page 5) “To assess this, we applied CODEC (214M raw read pairs) and Duplex Sequencing (305M) to WGS of the pilot genome NA12878 of the Genome in a Bottle Consortium [31].”

(page 5) “In a cost-benefit analysis, the cost of CODEC was 87 times lower than that of Duplex Sequencing while maintaining higher accuracy than standard WGS (**Fig. 2d**).”

(page 5) “To test germline mutation detection, we used an established caller (GATK HaplotypeCaller) instead of the single-fragment mutation caller and only the correct product data (see ‘Germline SNV and small indel calling in downsampled WGS’ in the Methods). When we compared CODEC and standard WGS of NA12878 at coverages ranging from 1x to 5x (0.6-3.0x correct product depth), CODEC showed 21-fold fewer false positives (FP) and 2-fold more false negatives (FN) for germline single nucleotide polymorphisms (SNPs) than standard WGS (**Fig. 3b**), which was mostly caused by byproducts. At 40x coverage (17x correct product depth), the FN rate was further reduced to 0.057 (0.026 using full 40x coverage including byproducts, Supplementary Table 2).”

(page 5) “To examine the potential for CODEC to detect low-abundance mutations, such as those arising from clonal hematopoiesis (CH), directly from single DNA duplexes, we analyzed 6x CODEC WGS (range 5.5-6.9x; duplex depth 0.47-1.02x) after Duplex-Repair and 6x standard WGS on buffy coat-derived germline DNA from 15 breast cancer patients.”

(page 6) “Targeting the single-fragment mutations detected by 4.3x (range 3-6x, duplex depth 0.47-0.83x) CODEC WGS on cfDNA of four healthy donors and four breast cancer patients, we performed deep Duplex Sequencing on matching buffy coat DNA to confirm their origin.”

(page 7) “With higher accuracy, 2x CODEC (range 1.4-2.4x, duplex depth 0.11-0.14x) had 83 times higher fraction of SNV validated by 60x standard WGS + Mutect2 than 2x standard WGS (**Fig. 4a**, Supplementary Fig. 17a), and increasing CODEC coverage improved sensitivity as expected (Supplementary Fig. 17b).”

(page 7) “To test how CODEC improves tumor mutation detection from liquid biopsy samples, we performed hybridization capture with personalized, tumor-informed probe panels [23] on both CODEC (median 24M raw read pairs, range 12M-30M) and Duplex Sequencing (median 52M, range 19-76M) libraries from four breast cancer patients’ cfDNA.”

3. The much lower sequencing requirement of CODEC vs Duplex Sequencing is unclear in view of the Supplementary Table data. For samples in figure S7 (cfDNA from BC and HD patients), the total read pair sequenced with CODEC are about half of the read pairs used with Duplex Sequencing. So, overall, it looks like CODEC, as presented, needs half the sequencing requirement of Duplex Seq, not 100-fold. This should be discussed or clarified.

Response: The actual number of reads assigned to each library does not reflect the reads required for analysis. For the samples in our prior Supplementary Fig. 7 (which is now Supplementary Fig. 8), we sequenced the libraries deeper than needed in order to reach a plateau in duplex recovery and thus estimate molecular diversity.

4. In Fig. 4h, could the authors please indicate whether the mutations identified were all the mutations targeted for the given tumors and, if not, what proportion was missed?

Response: Thank you for suggesting another helpful analysis that shows sensitivity of CODEC. Out of 581 SNV from four patients, CODEC detected 578 SNV which was 99.5%. We have added this analysis as the new Supplementary Table 3.

Updated text: (page 7) “Four patients with different tumor fractions showed up to 100-fold reduction on read pairs needed to detect the tracked mutations (**Fig. 4h**, Supplementary Table 3).”

5. This sentence is unclear: “Although single strand consensus (SSC) was more accurate than the no consensus and paired-end reads consensus (R1+R2), without a consensus of Watson and Crick strands, its residual mutation frequency error rate was still 100 234-fold higher than that of CODEC.” Do the authors mean that SSC has more error than CODEC? But that is irrelevant since the comparison should be at the duplex level, right?

Response: Yes, we wanted to demonstrate that a consensus of Watson and Crick strands is the key to higher sequencing accuracy, which may not be obvious to some readers. To clarify our point, we have restructured the sentence.

Updated text: (page 3) “As expected, a consensus of Watson and Crick strands was crucial for suppressing errors; although single strand consensus (SSC) was more accurate than the no consensus and paired-end reads consensus (R1+R2) (Supplementary Fig. 7 shows how each consensus works), residual mutation frequency of CODEC was 234-fold lower than that of SSC.”

In addition, the residual mutation frequency might not be necessarily error but low frequency biological mutations?

Response: Yes, we agree. In our last round of revision, we had changed all prior instances of ‘error rate’ to ‘residual mutation frequency’ as you had suggested. We had indicated these changes with strikethroughs in the manuscript file which may have been lost when the text was copied above. Nonetheless, we can assure you that we have fixed this in the paper as shown below.

Updated text: (page 3) “As expected, a consensus of Watson and Crick strands was crucial for suppressing errors; although single strand consensus (SSC) was more accurate than the no consensus and paired-end reads consensus (R1+R2) (Supplementary Fig. 7 shows how each consensus works), residual mutation frequency of CODEC was still 234-fold lower than that of SSC.”

Decision Letter, second revision:

25th Nov 2022

Dear Viktor,

I hope that you and your co-authors have had a relaxing Thanksgiving holiday!

Your Technical Report, "CODEC enables 'single duplex' sequencing" has now been seen by 2 of the original referees. You will see from their comments below that they consider your manuscript improved, but there is still one important question remaining. We remain interested in the possibility of publishing your study in Nature Genetics, but would like to consider your response to these concerns in the form of a revised manuscript before we make a final decision on publication.

Briefly, Reviewer #4 is now satisfied; they still think the presentation is confusing, but they're not sure there's a significantly better way of laying out your results.

Reviewer #3 appreciates most of your responses but still thinks that the question of how sensitive

CODEC is for detecting cancer somatic mutations is an important one that, currently, is not clearly presented. They specifically ask how much duplex sequencing is needed for CODEC to be as sensitive as deep (60x) WGS. We think this would be a very enlightening statistic that would be of interest to a broad range of users (not only cancer researchers), and ideally you would be able to supply the experimental data this Reviewer is after. However, we also acknowledge that this is a late round of review and there may be other reasons for this lack of data (e.g. you've used up all of the sequencing library or biosample), so we will not make it a condition for an eventual publication.

To guide the scope of the revisions, the editors discuss the referee reports in detail within the team, including with the chief editor, with a view to identifying key priorities that should be addressed in revision and sometimes overruling referee requests that are deemed beyond the scope of the current study. We hope that you will find the prioritized set of referee points to be useful when revising your study. Please do not hesitate to get in touch if you would like to discuss these issues further.

We therefore invite you to revise your manuscript taking into account all reviewer and editor comments. Please highlight all changes in the manuscript text file. At this stage we will need you to upload a copy of the manuscript in MS Word .docx or similar editable format.

*2) If you have not done so already please begin to revise your manuscript so that it conforms to our Technical Report format instructions, available [here](http://www.nature.com/ng/authors/article_types/index.html). Refer also to any guidelines provided in this letter.

[redacted]

Note: This URL links to your confidential home page and associated information about manuscripts you may have submitted, or that you are reviewing for us. If you wish to forward

this email to co-authors, please delete the link to your homepage.

We hope to receive your revised manuscript within four to eight weeks. If you cannot send it within this time, please let us know.

Sincerely,

Michael Fletcher, PhD
Senior Editor, Nature Genetics

ORCID: 0000-0003-1589-7087

Reviewers' Comments:

Reviewer #3:

Remarks to the Author:

The authors answered most of my questions. I fully understand that CODEC's 'positive predictive value (PPV)' is superior to general NGS. Therefore, using a low amount of CODEC sequencing (i.e., 1x or less), we can accurately detect the mutation density and the mutational signature. There are many CODEC applications, including NIPT and cfDNA. For these applications, the PPV is more important than the sensitivity.

However, in line with my previous comment (comment #1), the sensitivity of mutation detection is also essential for some purposes, i.e., primary cancer genome sequencing to sort out most (if not all) driver mutations. The state-of-the-art method is the deep (let's say 60x) whole-genome sequencing of cancer with its matched normal tissues. Can CODEC be applied for it? (Or do the authors think it is not a potential application of CODEC?) To this end, CODEC's high 'sensitivity' should be fully described in the main manuscript. However, the issue is not clearly described in the revised manuscript.

For example, on page 7;

"With higher accuracy, 2x CODEC (range 1.4-2.4x, duplex depth 0.11-0.14x) had 83 times higher fraction of SNV validated by 60x standard WGS + Mutect2 than 2x standard WGS"

However, it is insufficient because nobody conducts '2x standard WGS' for cancer genome analyses. We expect nothing from the '2x standard WGS'. The comparison may be technically fair but does not say anything about the 'utility' of CODEC in the real world.

In this manner, the following sentence, 'and increasing CODEC coverage improved sensitivity as expected (Supplementary Figure 17b).' is a meaningful description. I believe that 'the amount of improvement' should be explicitly shown by the authors, part of which is in Supplementary Fig. 17b. (Therefore, I think that Supplementary Fig. 17b should be displayed in the main figure.)

According to the figure, 3.7 duplex sequencing showed ~65% sensitivity, which increases to 94% when non-duplex reads were further combined (in this case, the PPV must be decreased). What I originally wanted to know was the duplex depth of CODEC to detect most (~95%?) clonal somatic mutations. It can be shown by increasing the CODEC sequencing depth from the tumor sample.

Reviewer #4:
Remarks to the Author:

I appreciate this new effort to clarify how the data was obtained for each figure. That said, I think the article is still confusing in some areas. For example, in the new paragraph in Page 3 it is not clearly defined what are CODEC byproducts. This sentence is confusing too: "At 40x coverage (17x correct product depth), the FN rate was further reduced to 0.057 (0.026 using full 40x coverage including byproducts, Supplementary Table 2)."

I think the main problem is that some of the analyses are performed with duplex data and some others are not. The authors have done an exhaustive effort fixing their comparisons to make sure they are consistent. I am not sure there is much more it can be done without completely rearranging the way the results are presented. The method is definitely not mature, but has potential.

Author Rebuttal, second revision:

Reviewers' Comments:

Reviewer #3:

Remarks to the Author:

The authors answered most of my questions. I fully understand that CODEC's 'positive predictive value (PPV)' is superior to general NGS. Therefore, using a low amount of CODEC sequencing (i.e., 1x or less),

we can accurately detect the mutation density and the mutational signature. There are many CODEC applications, including NIPT and cfDNA. For these applications, the PPV is more important than the sensitivity.

However, in line with my previous comment (comment #1), the sensitivity of mutation detection is also essential for some purposes, i.e., primary cancer genome sequencing to sort out most (if not all) driver mutations. The state-of-the-art method is the deep (let's say 60x) whole-genome sequencing of cancer with its matched normal tissues. Can CODEC be applied for it? (Or do the authors think it is not a potential application of CODEC?) To this end, CODEC's high 'sensitivity' should be fully described in the main manuscript. However, the issue is not clearly described in the revised manuscript.

Response: Thank you for this insightful comment. Yes, we believe that CODEC can be applied for deep (e.g., 60x) WGS of tumor and normal tissue. Yet, its highly accurate duplex reads are unlikely to provide higher sensitivity for the same high abundance mutations found in deep (e.g., 60x) standard WGS. This is because, for deep WGS, the sensitivity for high abundance mutations is driven more so by NGS depth than accuracy, and CODEC's duplex depth will be less than its overall depth. We have shown this in our new Fig. 4b in response to your comment below (Note: it is also possible that the "ground truth" of deep standard WGS is imperfect and that deep CODEC WGS could be more accurate than it, but we do not consider this here).

Where CODEC stands to shine is **detecting low abundance mutations at or below the noise floor** of deep (e.g., 60x) standard WGS. Conceptually, deep CODEC WGS stands to uncover more total mutations (including low frequency mutations below the noise floor of deep standard NGS) and thus, it may afford greater overall sensitivity. Yet, the challenge is two-fold: (1) the noise floor of deep standard WGS is not well defined (e.g., we know mutation detection becomes less reliable at lower VAFs but not to what degree), and (2) we lack a complete "ground truth" of all mutations in a tumor (e.g., simply using deep standard WGS as the "ground truth" to estimate the sensitivity of deep CODEC would be insufficient).

However, we appreciate the need for greater clarity about the performance of deeper WGS. We have thus followed your specific guidance below, adding **a new Fig. 4b** and several clarifications about CODEC's potential, as we presently know, for deep WGS. While fully addressing this important question is beyond the scope of this paper, we believe that our added analyses and clarifications set realistic expectations for what to expect from CODEC and where to focus further investigations into its performance.

For example, on page 7;

"With higher accuracy, 2x CODEC (range 1.4-2.4x, duplex depth 0.11-0.14x) had 83 times higher fraction of SNV validated by 60x standard WGS + Mutect2 than 2x standard WGS"

However, it is insufficient because nobody conducts '2x standard WGS' for cancer genome analyses. We expect nothing from the '2x standard WGS'. The comparison may be technically fair but does not say anything about the 'utility' of CODEC in the real world.

In this manner, the following sentence, 'and increasing CODEC coverage improved sensitivity as expected (Supplementary Figure 17b).' is a meaningful description. I believe that 'the amount of improvement' should be explicitly shown by the authors, part of which is in Supplementary Fig. 17b. (Therefore, I think that Supplementary Fig. 17b should be displayed in the main figure.)

Response: Thank you for your suggestion. We have moved the Supplementary Fig. 17b to be a main figure (new Fig. 4b) and further revised it in response to your next comment.

Updated text: (New Fig. 4b caption on page 7) "(b) Sensitivity for detecting clonal tumor mutations from deep CODEC WGS (3.7x duplex depth, 18x coverage) and corresponding positive predictive value (PPV) of mutations identified by CODEC. The data were analyzed by requiring either ≥ 1 or ≥ 2 duplexes bearing the same mutation. Standard NGS at 18x coverage paired with Mutect2 [34] achieved 98.6% sensitivity and 92.8% PPV. Theoretical numbers and projections (dashed) were calculated based on binomial models."

According to the figure, 3.7 duplex sequencing showed ~65% sensitivity, which increases to 94% when non-duplex reads were further combined (in this case, the PPV must be decreased). What I originally wanted to know was the duplex depth of CODEC to detect most (~95%?) clonal somatic mutations. It can be shown by increasing the CODEC sequencing depth from the tumor sample.

Response: Our old Supplementary Fig. 17b (new Fig. 4b) showed a clear trend of sensitivity vs duplex depth; thus, we reasoned that the duplex depth to reach a certain sensitivity can be simulated without further sequencing. Using binomial models (Jennings *et al.*, *J. Mol. Diagn.* 2017), we estimated that 8x duplex depth (resulting from 40x deduplicated coverage when using all reads) would be needed to detect 95% of clonal mutations with single CODEC's duplex reads. Yet, we also reasoned that single duplex reads may reveal mutations of varied abundances, whereas requiring ≥ 2 unique duplexes bearing a mutation would focus the analysis on higher abundance mutations. For this reason, we also explored how many duplexes

would be required to detect 95% of clonal mutations using ≥ 2 duplexes (which showed consistently high PPVs of 96.9%-97.9% across a 10-fold range of depths) and predicted that an even higher duplex depth of 10x (50x deduplicated coverage) would be needed.

For comparison, we downsampled 60x standard WGS to 18x and ran Mutect2. We found 98.6% of clonal mutations at 92.8% PPV. While performance may be inflated by this approach (i.e., sampling reads from the same standard NGS library used to define the “ground truth” at 60x), it provides a relevant point of comparison.

Our findings suggest that clonal mutation discovery is feasible with deep CODEC WGS; although expectedly, it is not more sensitive than standard NGS. We believe greater opportunity is to detect more total mutations including those below the noise floor of standard NGS, as suggested by high PPVs of duplex reads in deeper CODEC WGS (new Fig. 4b) and mutation validation in other samples (Fig. 4c). Yet, at present, we lack the complete “ground truth” of all tumor mutations to make this assessment.

Updated text: (Page 8) “Of note, deep CODEC WGS did not outperform deep (e.g., 60x) standard WGS in its sensitivity for high abundance (e.g., clonal) tumor mutations, which made sense as this is driven more by NGS depth than NGS accuracy. That said, we saw high positive predictive values (PPVs) of mutations detected in single duplex reads from deep CODEC WGS, suggesting that single CODEC reads are accurate enough to detect true tumor mutations. To further explore this, we performed targeted resequencing of mutations identified exclusively in single duplexes of 2x CODEC WGS and found more than a quarter to be validated as true mutations (**Fig. 4c**). The portion of mutations exclusively identified by CODEC showed VAF down to 0.042%.”

(Page 10) “As for whether CODEC could replace standard NGS for deep WGS of cancer and normal tissue, further investigation is required. On one hand, deep CODEC WGS does not presently show higher sensitivity for high abundance (e.g., clonal) mutations that are ordinarily detected in deep (e.g., 60x) standard NGS, but this was expected because (1) the sensitivity for clonal mutations is mostly driven by NGS depth as opposed to accuracy, and (2) CODEC’s duplex depth is currently lower than its full NGS depth. This is expected to improve, however, as CODEC’s duplex yield is further optimized, and we propose that mutation callers could be optimized to harness all CODEC reads, including non-duplex reads. On the other hand, we have found high PPVs for CODEC single duplex reads, suggesting that deep CODEC may be able to uncover more total mutations in a tumor biopsy such as those which are otherwise below the noise floor of deep, standard NGS. The challenge is that we currently lack a complete “ground truth” for all mutations in a tumor, making this difficult to evaluate. Further studies are thus needed to investigate whether deep CODEC WGS achieves higher overall sensitivity for tumor mutations over a wider and deeper range of allele frequencies.”

(Page 13)

“Examining clonal tumor mutation detection with CODEC WGS

We deeply sequenced a tumor biopsy (Fig. 4b) of 0.86 tumor purity using CODEC with 509M raw read pairs. The mean insert size was 131bp and duplication rate was 25%. The purity was estimated by ABSOLUTE [58] on 60x standard tumor WGS. We downsampled the data to 10%, 20%,..., 90% of total reads and called mutations using the single-fragment mutation caller. To strike a balance between sensitivity and specificity, we chose to use a Q30+Q30 cutoff at 50%. The rest of the filters were kept the same as described in the earlier section. From all somatic SNVs called from 60x WGS by Mutect2, we selected a set of SNVs (n=3,408) with cancer cell fraction (CCF, estimated by ABSOLUTE) ≥ 0.9 and within our high-complexity regions (2.3B) as clonal SNVs. When estimating the PPV, we used all SNVs (n=11,714) as the ground truth without restricting to CCF. For CODEC data, we called SSNVs with at least 1 or 2 mutant duplex support. The theoretical sensitivities were estimated by a binomial distribution with $p = 0.86 \times 0.5$, assuming all heterozygous mutations. The projection was through binomial distributions where p was estimated from the data, which we found to be lower than the ideal case.”

Reviewer #4:

Remarks to the Author:

I appreciate this new effort to clarify how the data was obtained for each figure. That said, I think the article is still confusing in some areas. For example, in the new paragraph in Page 3 it is not clearly defined what are CODEC byproducts. This sentence is confusing too: “At 40x coverage (17x correct product depth), the FN rate was further reduced to 0.057 (0.026 using full 40x coverage including byproducts, Supplementary Table 2).”

Response: Thank you for the comment. We added a reference to the CODEC byproducts defined in the Supplementary Fig. 4. We also slightly rephrased the sentence you mentioned on page 5.

Updated text: (Page 3) “Raw and deduplicated coverages include all CODEC byproduct (Supplementary Fig. 4).”

(Page 5) “At 40x coverage (17x correct product depth), the FN rate was further reduced to 0.057. The best FN (0.026) was achieved when also including non-duplex reads such as the byproducts (Supplementary Table 2).”

I think the main problem is that some of the analyses are performed with duplex data and some others are not. The authors have done an exhaustive effort fixing their comparisons to make sure they are consistent. I am not sure there is much more it can be done without completely rearranging the way the results are presented. The method is definitely not mature, but has potential.

Response: Thank you for your comments. Multiple types of analyses had to be performed considering the varied data types and applications. We have done our best to standardize the comparisons and provide the details needed for readers to reproduce our results and further extend CODEC across a variety of settings.

Decision Letter, third revision:

11th Jan 2023

Dear Viktor,

Thank you for submitting your revised manuscript "CODEC enables 'single duplex' sequencing" (NG-TR59556R2). It has now been seen by the original referee #3 and their comments are below. The reviewers find that the paper has improved in revision, and therefore we'll be happy in principle to publish it in Nature Genetics, pending minor revisions to satisfy the referees' final requests and to comply with our editorial and formatting guidelines.

Sincerely,

Michael Fletcher, PhD
Senior Editor, Nature Genetics

ORCID: 0000-0003-1589-7087

Reviewer #3 (Remarks to the Author):

No more comments

Final Decision Letter:

Pending Acceptance